# 1q amplification and *PHF19* expressing high-risk cells are associated with relapsed/ refractory multiple myeloma

Travis S. Johnson [1,2,3,4], Parvathi Sudha [5], Enze Liu[5], Nathan Becker[5], Sylvia Robertson[2], Patrick Blaney [6], Gareth Morgan [6], Vivek S. Chopra[7], Cedric Dos Santos[7], Michael Nixon[8], Kun Huang [1,3,4], Attaya Suvannasankha [5,9], Mohammad Abu Zaid [5], Rafat Abonour[5] & Brian A. Walker [4,5] ✉

Multiple Myeloma is an incurable plasma cell malignancy with a poor survival rate that is usually treated with immunomodulatory drugs (iMiDs) and proteosome inhibitors (PIs). The malignant plasma cells quickly become resistant to these agents causing relapse and uncontrolled growth of resistant clones. From whole genome sequencing (WGS) and RNA sequencing (RNA-seq) studies, different high-risk translocation, copy number, mutational, and transcriptional markers can be identified. One of these markers, *PHF19*, epigenetically regulates cell cycle and other processes and is already studied using RNA-seq. In this study, we generate a large (325,025 cells and 49 patients) single cell multi-omic dataset and jointly quantify ATAC- and RNA-seq for each cell and matched genomic profiles for each patient. We identify an association between one plasma cell subtype with myeloma progression that we call relapsed/refractory plasma cells (RRPCs). These cells are associated with chromosome 1q alterations, TP53 mutations, and higher expression of *PHF19*. We also identify downstream regulation of cell cycle inhibitors in these cells, possible regulation by the transcription factor (TF) PBX1 on chromosome 1q, and determine that *PHF19* may be acting primarily through this subset of cells.

Myeloma is a plasma cell malignancy with a poor survival rate that in current years has seen improved prognosis due to advances in treatment[1]. Disease progression and more generally disease risk, are defined by molecular subtypes of myeloma[2]. Currently there are multiple genomic markers that impact patient survival and likelihood to relapse[3,4] such as t(4;14)[5], Del17p[3,6], or Gain/Amp1q[7,8] which have been long associated with worse prognosis. Additional copies of chromosome 1 (Gain/Amp1q) are the most common chromosomal abnormalities in MM. Recently, Gain/Amp1q and especially Amp1q have emerged as a poor risk factor including risk of relapse with the current frontline therapies for myeloma[9]. Despite this attention, the mechanism through which Gain/Amp1q affects prognosis and response to therapy is still not known. Some hypotheses for the Gain/Amp1q mechanism include increased expression of *CKS1B* affecting

[1]Department of Biostatistics and Health Data Science, School of Medicine, Indiana University, Indianapolis, IN, USA. [2]Indiana Biosciences Research Institute, Indianapolis, IN, USA. [3]Melvin and Bren Simon Comprehensive Cancer Center, Experimental and Developmental Therapeutics, School of Medicine, Indiana University, Indianapolis, IN, USA. [4]Center for Computational Biology and Bioinformatics, School of Medicine, Indiana University, Indianapolis, IN, USA. [5]Melvin and Bren Simon Comprehensive Cancer Center, Division of Hematology and Oncology, School of Medicine, Indiana University, Indianapolis, IN, USA. [6]Perlmutter Cancer Center, Langone Health, New York University, New York, NY, USA. [7]Genentech Inc., South San Francisco, CA, USA. [8]Roche Inc., Indianapolis, IN, USA. [9]Roudebush VAMC, Indianapolis, IN, USA. ✉e-mail: bw75@iu.edu

*SKP2* and *KIP27*[10] or more recently the transcription factor (TF), *PBX1*, affecting *FOXM1*[11]. For these reasons there is still a need to understand the mechanisms through which Gain/Amp1q leads to poor prognosis.

Besides translocations and copy number variations that affect myeloma disease progression, there are many epigenetic regulators that affect chromatin accessibility and result in altered protein expression and eventually progression[12]. PHD finger protein 19 (PHF19) is an epigenetic regulator that has recently seen much interest in the myeloma field due to its relationship with poor patient outcomes[13]. It has been shown that PHF19 can affect cell cycle by interacting with the polycomb repressive complex 2 (PRC2) to induce trimethylation of histone H3 lysine 27 (H3K27me3)[14]. Over-expression of *PHF19* has been shown to increase H3K27me3 at cell cycle inhibitor and JAK-STAT loci in the genome[14]. Though these regulatory relationships have been evaluated using cell line experiments and using patient-level omics datasets there has yet to be an evaluation of these mechanisms at the single-cell level.

Recently, single cell RNA sequencing datasets have been generated from cohorts of myeloma patients. Initially, these datasets showed that there were key differences between asymptomatic multiple myeloma patients and symptomatic multiple myeloma patients[15]. Both in patients with asymptomatic multiple myeloma and patients post treatment with minimum residual disease there were detectable levels of rare tumor cells with characteristics of active myeloma[15]. Subsequently, relapsed and refractory patients after bortezomib treatment acquired new resistance mechanisms including hypoxia tolerance, protein folding, and mitochondria respiration[16]. Aside from noticeable changes in the tumor cell fraction that improve resistance to treatment and more aggressive proliferation, the immune microenvironment is compromised in earlier precursors to multiple myeloma[17]. All of these studies show that the complex interplay between the tumor and immune microenvironments are largely dysregulated even in early stages of disease. These changes can sometimes be attributed to epigenetic regulatory mechanisms[12].

In this study, we use a large single-cell multiomic dataset to show how cytogenetic changes affect the progression of myeloma. Using chromatin accessibility paired with gene expression we are able to recapitulate epigenetic regulation of the cell cycle in human samples at the single-cell level. We are able to link these epigenetic relationships back to cytogenetic events uncovering new mechanisms through which genomic events lead to increased cell proliferation.

## Results

### Integrated clustering identifies cells associated with relapse and high-risk markers

CD138+ single cells were assayed by multiome sequencing (10X genomics) from 49 patients with SMM ($n = 10$), NDMM ($n = 22$), and RRMM ($n = 17$) (Fig. 1, Table 1). Samples initially had an average of 9484 cells. Removal of low-quality cells and non-plasma cells resulted in 6819 cells per sample for downstream analyses (Supplementary Fig. 1, Supplementary Tables 1 and 2). Based on our analysis pipeline to

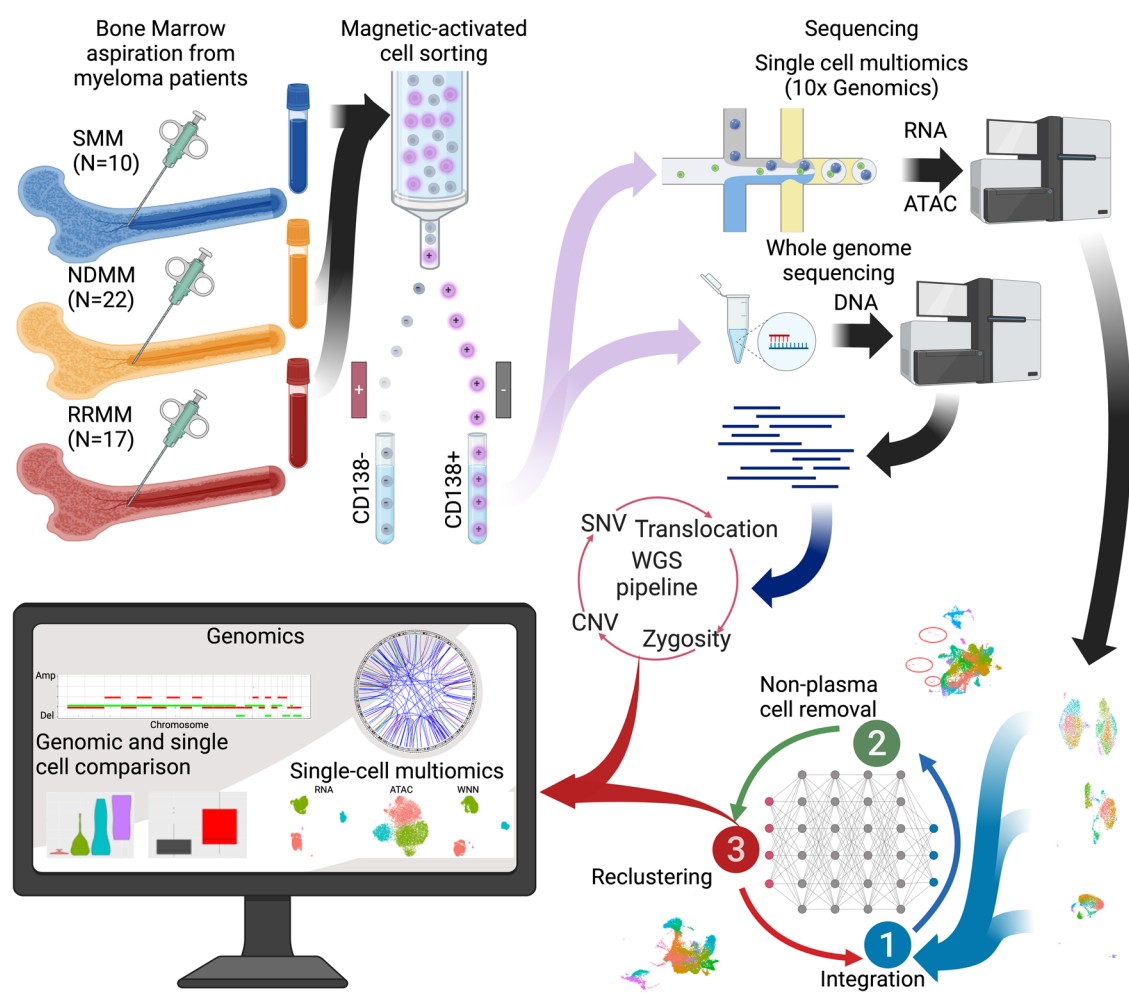

**Fig. 1 | Overview of the samples collection, processing, and our analysis pipelines for WGS and single cell multiomics (created by BioRender).** Samples were collected from SMM, NDMM, and RRMM patients and sorted into the CD138+ fraction. This fraction was then split and sent for bulk WGS and single cell multiomic (RNA + ATAC) sequencing. These cells and patient WGS profiles then underwent our computational pipeline.

**Table 1 | Patient demographics**

| Diagnosis | SMM | NDMM | RRMM | All |
|---|---|---|---|---|
| *N* = | 10 | 22 | 17 | 49 |
| Age > 65 (%) | 5 (50%) | 14 (64%) | 11 (65%) | 30 (61%) |
| Sex (% male) | 4 (40%) | 11 (50%) | 13 (76%) | 21 (43%) |
| Translocations | | | | |
| t(4;14) (n/%) | 2 (20%) | 1 (5%) | 2 (12%) | 5 (10%) |
| t(6;14) | 0 (0%) | 1 (5%) | 0 (0%) | 1 (2%) |
| t(11;14) | 1 (10%) | 5 (23%) | 4 (24%) | 10 (20%) |
| t(14;16) | 0 (0%) | 0 (0%) | 1 (6%) | 1 (2%) |
| t(14;20) | 0 (0%) | 1 (5%) | 1 (6%) | 2 (4%) |
| Hyperdiploidy | 7 (70%) | 15 (68%) | 8 (47%) | 30 (61%) |
| CNV | | | | |
| Del(*CDKN2C*)/1p | 0 (0%) | 6 (27%) | 3 (18%) | 9 (18%) |
| Gain(*CKS1B*)/1q | 6 (60%) | 8 (36%) | 6 (35%) | 29 (59%) |
| Gmp(*CKS1B*)/1q | 0 (0%) | 4 (18%) | 5 (29%) | 9 (18%) |
| Del(*RB1*)/13q | 6 (60%) | 10 (45%) | 11 (65%) | 27 (55%) |
| Del(*TP53*)/17p | 0 (0%) | 2 (9%) | 5 (29%) | 7 (14%) |
| Mutations[a] | | | | |
| *TENT5C* | 0 (0%) | 1 (5%) | 1 (7%) | 2 (5%) |
| *TP53* | 0 (0%) | 2 (10%) | 6 (40%) | 8 (18%) |
| *BRAF* | 0 (0%) | 1 (5%) | 2 (13%) | 3 (7%) |
| *NRAS* | 1 (11%) | 4 (20%) | 3 (20%) | 8 (18%) |
| *KRAS* | 2 (22%) | 5 (25%) | 2 (13%) | 9 (20%) |

The following abbreviations were used in the table: smoldering multiple myeloma (SMM), newly diagnosed multiple myeloma (NDMM), relapsed or refractory multiple myeloma (RRMM), copy number variation (CNV), loss of heterozygosity (LOH), mutation (Mut), and hyperdiploidy (HRD).
[a]Note that undetermined samples were excluded from the percentage calculation resulting in SMM (*N* = 9), NDMM (*N* = 20), and RRMM (*N* = 15).

integrate all patient samples, 325,025 high quality myeloma cells were retained for further study. These cells were used to impute translocations: t(4;14), t(6;14), t(11;14), t(14;16), and t(14;20) (Supplementary Fig. 2), copy number variants (CNVs): 1p, 1q, 13q, and 17p (Supplementary Fig. 3), and hyperdiploidy (HRD)(Supplementary Fig. 4) in the five patient samples missing WGS (Supplementary Table 3). Of the patients, 19 had a translocation that could be stratified into the primary translocation groups based on marker gene expression (Fig. 2A).

Cells were integrated and formed 25 distinct clusters (Fig. 2B) of myeloma cells with unique transcriptome profiles that frequently associated with stage of progression (Fig. 2C, Supplementary Fig. 5) and genomic events (Fig. 2D, Supplementary Data 1). Using the previously defined clusters (Fig. 2B), we determined if samples in any disease stage, cytogenetic subgroup, or genetic subgroup (Supplementary Table 4) are enriched for cells in any particular cluster (Fig. 2D). We determined that clusters 11, 20, 22, and 23 were associated with later stages of myeloma progression (Fig. 2B–E) such that all of these clusters had significantly greater proportions in RRMM compared to SMM ($P < 0.001$ and Benjamini-Hochberg false discovery rate (FDR) = 0.003, $P < 0.001$ and FDR = 0.003, $P = 0.031$ and FDR = 0.24, and $P = 0.032$ and FDR = 0.25, respectively, Supplementary Data 1). The largest of these clusters, cluster 11, increased in proportion from SMM to NDMM (1% to 3%, $P = 0.008$ and FDR = 0.13, Fig. 2F, Supplementary Data 1), and NDMM to RRMM (3–8%, $P < 0.001$ and FDR = 0.03, Fig. 2F, Supplementary Data 1). Due to their greater enrichment in RRMM patients in comparison to earlier stages of myeloma we denote clusters 11, 20, 22, and 23 as relapse/refractory plasma cells (RRPCs) or $RRPC_{11}$, $RRPC_{20}$, $RRPC_{22}$, and $RRPC_{23}$ for individual RRPC clusters (Table 2).

Aside from a clear association of RRPCs with myeloma progression, there were also associations between samples with high-risk genomic events and their proportion of RRPCs (Fig. 2D, Supplementary Data 1). The proportion of all RRPCs was associated with 1q copy number such that samples with a normal copy number of 1q had a significantly lower proportion of RRPCs compared to Gain/Amp1q samples (4% vs. 8%, $P = 0.019$). Patients with Gain/Amp1q (defined by WGS) had significantly more cells in cluster $RRPC_{11}$ than those with a normal copy number of 1q (0.05 vs. 0.03, $P = 0.030$ and FDR = 0.24; Fig. 2G, Supplementary Data 1) and those with Amp1q had significantly more $RRPC_{11}$ than those with Gain1q (4% vs. 9%, $P = 0.014$; Fig. 2G). In addition, samples with a Mut(*TP53*) had a significantly greater proportion of $RRPC_{11}$ than samples without Mut(*TP53*) (0.08 vs. 0.03, $P = 0.025$ and FDR = 0.22; Fig. 2H, Supplementary Data 1). When the $RRPC_{11}$ fraction of cells was screened for Mut(*TP53*) in the scATAC-seq reads (i.e., variants originally identified by WGS), we were able to identify those variants in 3/5 of the samples with read coverage of the WGS variant loci (Supplementary Fig. 6) and those $RRPC_{11}$ cells had an average Mut(*TP53*) variant allele frequency (VAF) of 55% compared to 32% in all other cells (Supplementary Table 5). There was also an additive effect between Mut(*TP53*) status and Gain/Amp1q such that the number of these alterations was correlated with $RRPC_{11}$ proportion (Pearson Correlation Coefficient (PCC) = 0.47, $P = 0.001$, Fig. 2I) and the $RRPC_{11}$ proportion in patients with both Gain/Amp1q and Mut(*TP53*) was equal to the $RRPC_{11}$ proportion in patients with Gain/Amp1q added to the $RRPC_{11}$ proportion in patients with Mut(*TP53*) accounting for the baseline $RRPC_{11}$ proportion in patients without either Gain/Amp1q or Mut(*TP53*) (Fig. 2I). Additionally, Del12p(*CDKN1B*) ($P = 0.027$, FDR = 0.227), *NRAS* mutations ($P = 0.039$, FDR = 0.276), and *HUWE1* mutations ($P = 0.048$, FDR = 0.289) were also associated with $RRPC_{11}$. Although clusters $RRPC_{11}$, $RRPC_{20}$, $RRPC_{22}$, and $RRPC_{23}$ were all significantly associated with Gain/Amp1q (Fig. 2D, Table 3), only $RRPC_{11}$, $RRPC_{20}$, and $RRPC_{22}$ were associated with Amp1q (Fig. 2D) and adjacently clustered to each other (Fig. 2B, C). RRPC clusters were not enriched for other high-risk markers such as Del(*CDKN2C*), t(4;14), or t(14;16). $RRPC_{11}$ was also the most prevalent of the RRPC clusters. All of these factors make RRPCs and more specifically $RRPC_{11}$ an important group of cells for further study.

## RRPCs have increased expression of the epigenetic modifier *PHF19* and proliferative genes

To better understand the function of RRPCs, marker genes were identified using differential gene expression analysis comparing each RRPC cluster to all other clusters. For $RRPC_{11}$, this identified 8232 up-regulated and 1590 down-regulated differentially expressed gene (DEGs) (Supplementary Data 2) and many were also differentially expressed in both $RRPC_{20}$ and $RRPC_{22}$ (1247 up-regulated and 191 down-regulated common genes, Supplementary Data 3, 4). $RRPC_{11,20,22}$ all had similar expression profiles (Fig. 3A). Notably, $RRPC_{11,20,22}$ had greater expression of *PHF19* than other clusters ($RRPC_{11}$: $Log_2FC = 2.43$, $P < 0.001$ and FDR < 0.001; $RRPC_{20}$: $Log_2FC = 3.08$, $P < 0.001$ and FDR < 0.001; $RRPC_{22}$: $Log_2FC = 2.77$, $P < 0.001$ and FDR < 0.001, Fig. 3A, B), an epigenetic modifier whose expression is highly predictive of poor prognosis in myeloma patients[13]. One of the main markers for proliferation, *MKI67* is also highly expressed in $RRPC_{11,20,22}$ ($RRPC_{11}$: $Log_2FC = 3.78$, $P < 0.001$ and FDR < 0.001; $RRPC_{20}$: $Log_2FC = 4.37$, $P < 0.001$ and FDR < 0.001; $RRPC_{22}$: $Log_2FC = 3.80$, $P < 0.001$ and FDR < 0.001, Fig. 3A, B) and co-expressed with *PHF19* (PCC = 0.37, $P < 0.001$, Fig. 3C). This correlation between *MKI67* and *PHF19* expression can also be identified in NDMM patients from the MMRF CoMMpass study (Supplementary Fig. 7, Supplementary Tables 6 and 7). In the CoMMpass NDMM cohort, *MKI67* and *PHF19* expression ($Log_2TPM$) were highly correlated (PCC = 0.70, $P < 0.001$ and FDR < 0.001, Supplementary Table 6). In the CoMMpass NDMM cohort Amp1q, Gain1q, and Normal 1q subpopulation, their PCC values were also high (PCC = 0.83, PCC = 0.65, PCC = 0.70, $P < 0.001$ and FDR < 0.001, $P < 0.001$ and FDR < 0.001, $P < 0.001$

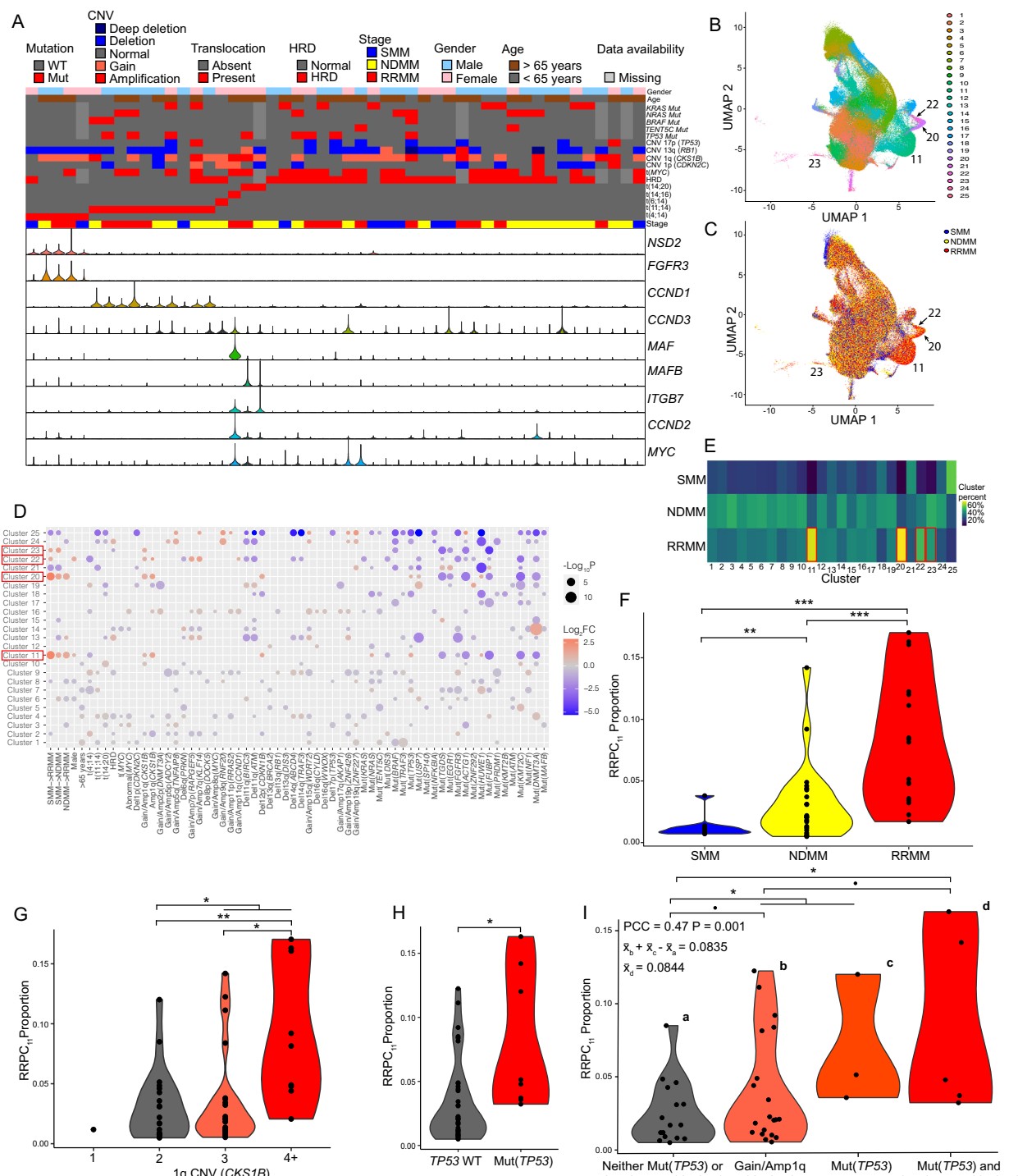

**Fig. 2 | Characterization and integration of plasma cell clusters. A** Canonical translocations, mutations, and copy number alteration marker gene expression in each patient. **B** Final clusters determined from the dataset integration of 49 samples. **C** Diagnosis of the patient from which each cell was derived overlaid on to the integrated clusters. **D** Significant associations between clusters and clinical/genomic covariates (one-sided t-tests). **E** Percentage of each cluster from SMM, NDMM, and RRMM patients. RRPC clusters are boxed in red (**D**, **E**). **F** Proportion of RRPCs in patients stratified by diagnosis (one-sided t-test p-values: NDMMvSMM $P = 8.26E-3$,

RRMMvNDMM $P = 8.18E-4$, RRMMvSMM $P = 2.17E-5$). **G** Proportion of $RRPC_{11}$ in patients stratified by 1q copy number (one-sided t-test p-values: 2v3+ $P = 2.96E-2$, 2v4 $P = 7.49E-3$, 3v4+ $P = 1.40E-3$). **H** Proportion of $RRPC_{11}$ in patients stratified by *TP53* mutation status (one-sided t-test p-value: WTvMut $P = 2.45E-2$). **I** Proportion of $RRPC_{11}$ in patients with Gain/Amp1q, *TP53* mutations, or both (one-sided t-test p-values: WTv1q $P = 8.11E-2$, WTv[1q or Mut(*TP53*)] $P = 3.50E-2$, WTv[1q and Mut(*TP53*)] $P = 4.85E-2$, 1qv[1q and Mut(*TP53*)] $P = 9.19E-2$). Significance levels are: $P < 0.1$: (·), $P < 0.05$: *, $P < 0.01$ (**), $P < 0.001$ (***).

and FDR < 0.001, respectively, Supplementary Table 6). Both MKI67 and PHF19 showed significantly overexpression in Amp1q vs. normal 1q ($Log_2FC = 0.49$, $Log_2FC = 0.52$, $P < 0.001$ and FDR < 0.001, $P = 0.002$ and FDR = 0.003, respectively, Supplementary Table 7). Furthermore,

we see enrichment for cell-cycle related processes in $RRPC_{11}$ (Fig. 3D, Supplementary Fig. 8A), $RRPC_{20}$ (Fig. 3E, Supplementary Fig. 8A), and $RRPC_{22}$ (Fig. 3F, Supplementary Fig. 8A). All RRPC clusters DEGs and differentially accessible chromatin (DACs) (|LogFC|>0.4, $P < 0.05$) were

also enriched (Gene Set Enrichment Analysis (GSEA) FDR < 0.25) for cancer Hallmarks like E2F targets, G2M checkpoint, and immune signaling pathways (Supplementary Tables 8–10). *PHF19* has been shown to negatively affect the expression of cell cycle inhibitors, therefore promoting proliferation[14]. Taking into consideration the association of RRPC clusters with Gain/Amp1q, *TP53* mutations, and increased

expression of *PHF19* (Supplementary Fig. 8B–D), RRPC clusters and especially $RRPC_{11,20,22}$ should be considered a high-risk subset of cells.

## RRPCs have reduced expression of cell-cycle inhibitors associated with loss of chromatin accessibility

Given that PHF19 is known to regulate negative regulators of cell cycle[14] we analyzed both the scRNA-seq and scATAC-seq data for changes in cell cycle and proliferation genes[18]. Compared to other clusters, proliferative index genes had increased chromatin accessibility (Fig. 4A) and increased expression (Fig. 4B) in the $RRPC_{11}$ cluster. In $RRPC_{11}$, *PHF19* was significantly up-regulated ($Log_2FC = 2.43$, $P < 0.001$, Fig. 4B), *CDKN1C* had reduced chromatin accessibility ($Log_2FC = -0.54$, $P < 0.001$, Fig. 4A) and reduced expression ($Log_2FC = -1.08$, $P < 0.001$, Fig. 4B), and *CDK4* had increased expression ($Log_2FC = 1.02$, $P < 0.001$, Fig. 4B) compared to all other clusters. Since, previous studies have already established epigenetic regulatory mechanisms of *CDKN1C* by PHF19[14] and *CDKN1C* is contained within a repressed polycomb regulatory region (Supplementary Fig. 9A)[19], these results demonstrate that a subset of proliferative cells with high *PHF19* expression, found primarily in relapsed or refractory patients, also likely epigenetically downregulate *CDKN1C* (Fig. 4A, B).

To further verify these patterns, data from cell lines where *PHF19* was knocked-down and subsequently rescued was analyzed[14]. A significant overlap between genes that were up-regulated from the *PHF19* knockdown experiments and genes downregulated in both the ATAC-seq and RNA-seq from $RRPC_{11}$ was found ($OR = 2.24$, $P = 0.010$, Fig. 4C). Similarly, the genes that were down-regulated by *PHF19* knockdown had significant overlap with genes found to be upregulated in both ATAC-seq and RNA-seq from $RRPC_{11}$ ($OR = 3.30$, $P < 0.001$, Fig. 4D). In

**Table 2 | Simplified table categorizing plasma cell clusters stratified by normal plasma cells (NPCs), hyperdiploid plasma cells (HDPCs), relapsed/refractory plasma cells (RRPCs), translocation plasma cells (TPCs), copy number alteration plasma cells (CNPCs), mutation plasma cells (MPCs), and undefined plasma cells**

| | |
|---|---|
| Normal plasma cells (NPCs) | 18 |
| Hyperdiploid plasma cells (HDPCs) | 1, 13, 24 |
| Relapsed/refractory plasma cells (RRPCs) | 11, 20, 22, 23 |
| t(4;14) plasma cells (TPCs) | 2, 9, 19 |
| t(11;14) plasma cells (TPCs) | 4, 7 |
| t(14;20) plasma cells (TPCs) | 10 |
| Del(*CDKN2A*) plasma cells (CNPCs) | 6 |
| Del(*PRKN*) plasma cells (CNPCs) | 14 |
| Gain/Amp(*ZNF227*) plasma cells (CNPCs) | 21, 25 |
| Abnormal(*MYC*) plasma cells (MPCs) | 3, 16 |
| Mut(*KMT2C*) plasma cells (MPCs) | 5 |
| Mut(*TGDS*) plasma cells (MPCs) | 15 |
| Undefined | 8, 12, 17 |

**Table 3 | Table with the defining genetic characteristics of plasma cell clusters (one-sided *t* test *P*<0.05 and FDR<0.25)**

| Cluster 18 (NPCs) | None |
|---|---|
| Cluster 1 (HDPC$_1$) | HRD, Gain/Amp(*CRBN*), Gain/Amp(*TNFAIP8*), Gain/Amp(*CDKN2A*), Gain/Amp(*WDR72*), Gain/Amp(*BLM*), Gain/Amp(*ZNF227*), Mut(*TENT5C*), Mut(*TRAF3*) |
| Cluster 13 (HDPC$_{13}$) | HRD, Gain/Amp(*RAPGEF5*), Gain/Amp(*KLF14*), Gain/Amp(*ATM*), Gain/Amp(*WDR72*), Gain/Amp(*BLM*), Gain/Amp(*ZNF426*) |
| Cluster 24 (HDPC$_{24}$) | HRD, Gain/Amp(*ADCY2*), Gain/Amp(*TNFAIP8*), Gain/Amp(*CDKN2A*), Amp(*CDKN2A*), Gain/Amp(*RNF20*), Gain/Amp(-*TRAF2*), Amp(*TRAF2*), Gain/Amp(*WDR72*), Gain/Amp(*BLM*), Gain/Amp(*ZNF227*) |
| Cluster 11 (RRPC$_{11}$) | Gain/Amp(*CKS1B*), Amp(*CKS1B*), Del(*CDKN1B*), Mut(*TP53*) |
| Cluster 20 (RRPC$_{20}$) | Gain/Amp(*CKS1B*), Amp(*CKS1B*), Del(*CDKN1B*) |
| Cluster 22 (RRPC$_{22}$) | Gain/Amp(*CKS1B*), Gain/Amp(*RAPGEF5*), Gain/Amp(*KLF14*) |
| Cluster 23 (RRPC$_{23}$) | Gain/Amp(*CKS1B*) |
| Cluster 2 (TPC$_2$) | t(4;14), Del(*CDKN2A*), Mut(*DIS3*), Mut(*FGFR3*) |
| Cluster 9 (TPC$_9$) | t(4;14), Del(*ABCD4*), Del(*TRAF3*), Mut(*TRAF3*), Mut(*FGFR3*), Mut(*NF1*) |
| Cluster 19 (TPC$_{19}$) | t(4;14), Del(*BIRC3*), Del(*ATM*), Mut(*USP7*), Mut(*ZNF292*) |
| Cluster 4 (TPC$_4$) | t(11;14), t(14;20), Del(*PRKN*), Del(*ABCD4*), Del(*TRAF3*), Mut(*FUBP1*), Mut(*DNMT3A*) |
| Cluster 7 (TPC$_7$) | t(11;14), Mut(*NRAS*), Mut(*TP53*), Mut(*HUWE1*) |
| Cluster 10 (TPC$_{10}$) | t(14;20), Gain/Amp(*CKS1B*), Mut(*KMT2C*) |
| Cluster 6 (CNPC$_6$) | Del(*CDKN2A*), Del(*BIRC3*), Mut(*USP7*) |
| Cluster 14 (CNPC$_{14}$) | Del(*PRKN*), Mut(*DNMT3A*) |
| Cluster 21 (CNPC$_{21}$) | Gain/Amp(*ZNF227*), Mut(*TGDS*), Mut(*EGR1*) |
| Cluster 25 (CNPC$_{25}$) | Gain/Amp(*RNF20*), Gain/Amp(*TRAF2*), Gain/Amp(*ZNF227*) |
| Cluster 3 (MPC$_3$) | Abnormal(*MYC*), Gain/Amp(*TNFAIP8*) |
| Cluster 16 (MPC$_{16}$) | Abnormal(*MYC*), Gain/Amp(*DNMT3A*), Amp(*DNMT3A*), Gain/Amp(*CRBN*), Amp(*CRBN*), Gain/Amp(*FGFR3*), Amp(*FGFR3*), Amp(*ADCY2*), Amp(*TNFAIP8*), Amp(*TNXB*), Amp(*RAPGEF5*), Gain/Amp(*KLF14*), Amp(*KLF14*), Amp(*CDKN2A*), Amp(*RNF20*), Amp(*TRAF2*), Gain/Amp(*RRAS2*), Amp(*RRAS2*), Gain/Amp(*CCND1*), Amp(*CCND1*), Amp(*ATM*), Gain/Amp(*CDKN1B*), Amp(*CDKN1B*), Gain/Amp(*ABCD4*), Gain/Amp(*TRAF3*), Gain/Amp(*WDR72*), Amp(*WDR72*), Gain/Amp(*BLM*), Amp(*BLM*), Del(*WWOX*), Gain/Amp(*AKAP1*), Amp(*AKAP1*), Gain/Amp(*ZNF426*), Amp(*ZNF426*), Amp(*ZNF227*), Gain/Amp(*SON*), Amp(*SON*) |
| Cluster 5 (MPC$_5$) | Mut(*KMT2C*) |
| Cluster 15 (MPC$_{15}$) | Mut(*TGDS*) |

The undefined plasma cells are excluded from the table

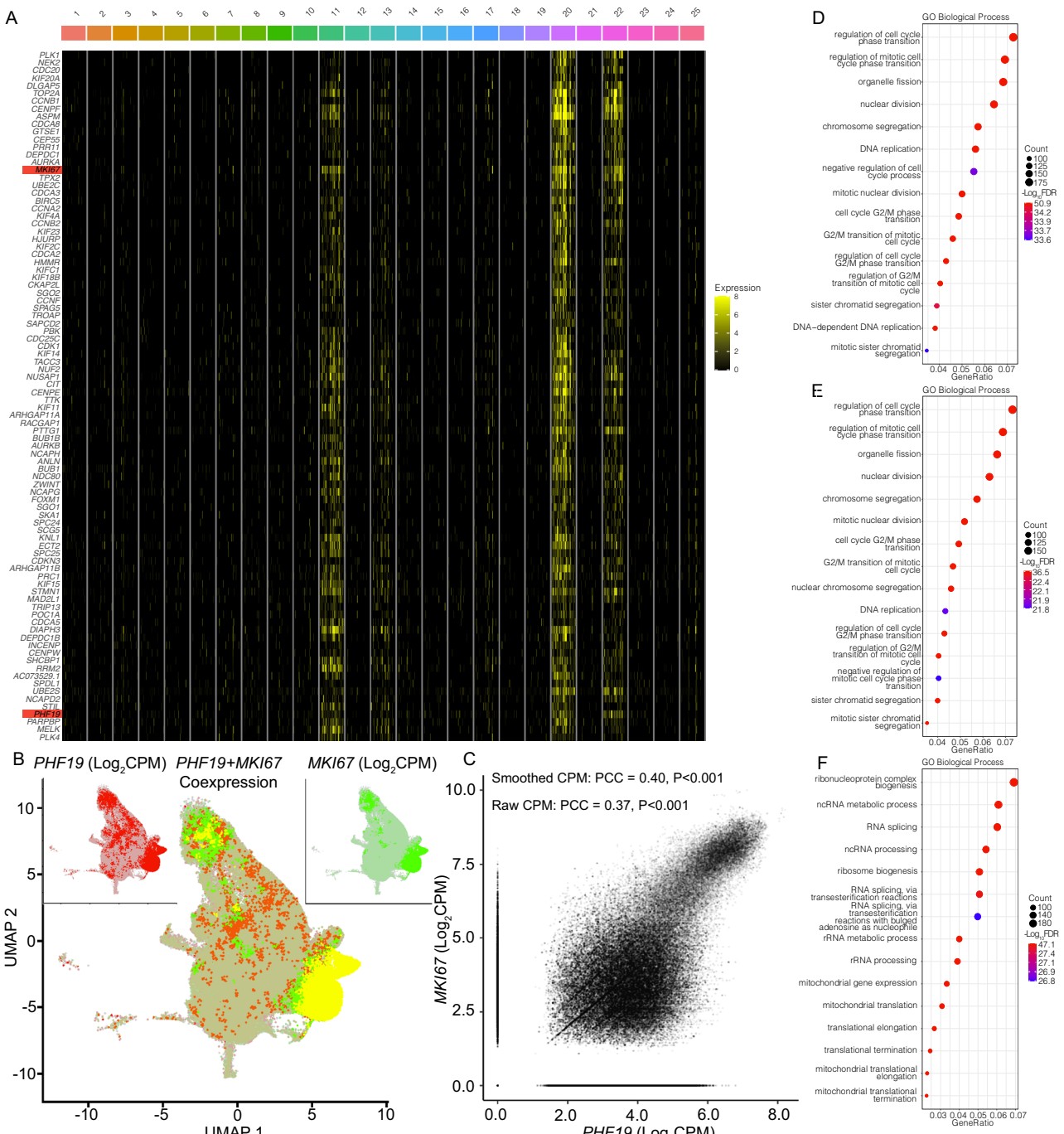

**Fig. 3 | Differential expression for the top intersecting RRPC$_{11}$, RRPC$_{20}$, and RRPC$_{22}$ markers.** **A** Heatmap of the top 90 up-regulated DEGs common between RRPC$_{11}$, RRPC$_{20}$, and RRPC$_{22}$. The DEGs were sorted by their rank sum across RRPC$_{11}$, RRPC$_{20}$, and RRPC$_{22}$. **B** RRPC clusters coexpress *PHF19* and *MKI67*. **C** *PHF19* and *MKI67* expression are correlated in their expression (PCC $P \approx 0.00$). Note that expression values were smoothed using 20 nearest neighbors for visualization purposes in **B** and **C**. **D**–**F** Top 15 significant (one-sided Fisher's exact test and BH-corrected) GO Biological Process for RRPC$_{11}$ DEGs (FDRs top to bottom: 2.38E−44, 3.66E−36, 6.61E−37, 2.13E−41, 1.25E−51, 2.32E−34, 3.41E−36, 3.07E−37, 4.38E−36, 1.04E−38, 4.02E−37, 5.41E−35, 1.03E−45, 2.71E−34) (**D**), RRPC$_{20}$ DEGs (FDRs top to bottom: 3.38E−37, 9.82E−37, 6.19E−28, 2.39E−29, 8.27E−35, 8.23E−33, 3.72E−32, 3.24E−31, 1.85E−27, 1.60E−22, 5.37E−32, 8.35E−31, 1.73E−22, 8.04E−31, 7.20E−30) (**E**), and RRPC$_{22}$ DEGs (FDRs top to bottom: 8.48E−48, 2.02E−30, 7.17E−31, 1.33E−32, 3.38E−43, 1.21E−28, 1.77E−27, 5.25E−35, 3.55E−35, 1.44E−38, 5.54E−43, 2.89E−30, 7.93E−35, 5.94E−38, 4.44−36) (**F**).

contrast neither of the opposite comparisons were significant. Genes that were up-regulated from the *PHF19* knockdown experiments and genes upregulated in both the ATAC-seq and RNA-seq from RRPC$_{11}$ had little overlap (OR = 0.76, *P* = 0.33). Genes that were down-regulated by *PHF19* knockdown had insignificant overlap with genes found to be downregulated in both ATAC-seq and RNA-seq from RRPC$_{11}$ (OR = 0.18,

*P* = 0.067). Upon further examination of the genes which were up-regulated in both ATAC-seq and RNA-seq, three were also in a known myeloma proliferation signature[18], namely: *NEK2*, *AURKB*, and *CCNB2* (Fig. 4E). These genes also represent potential therapeutic targets for relapsed/refractory patients who have had multiple failed treatments that would target the high risk RRPC$_{11}$ cluster.

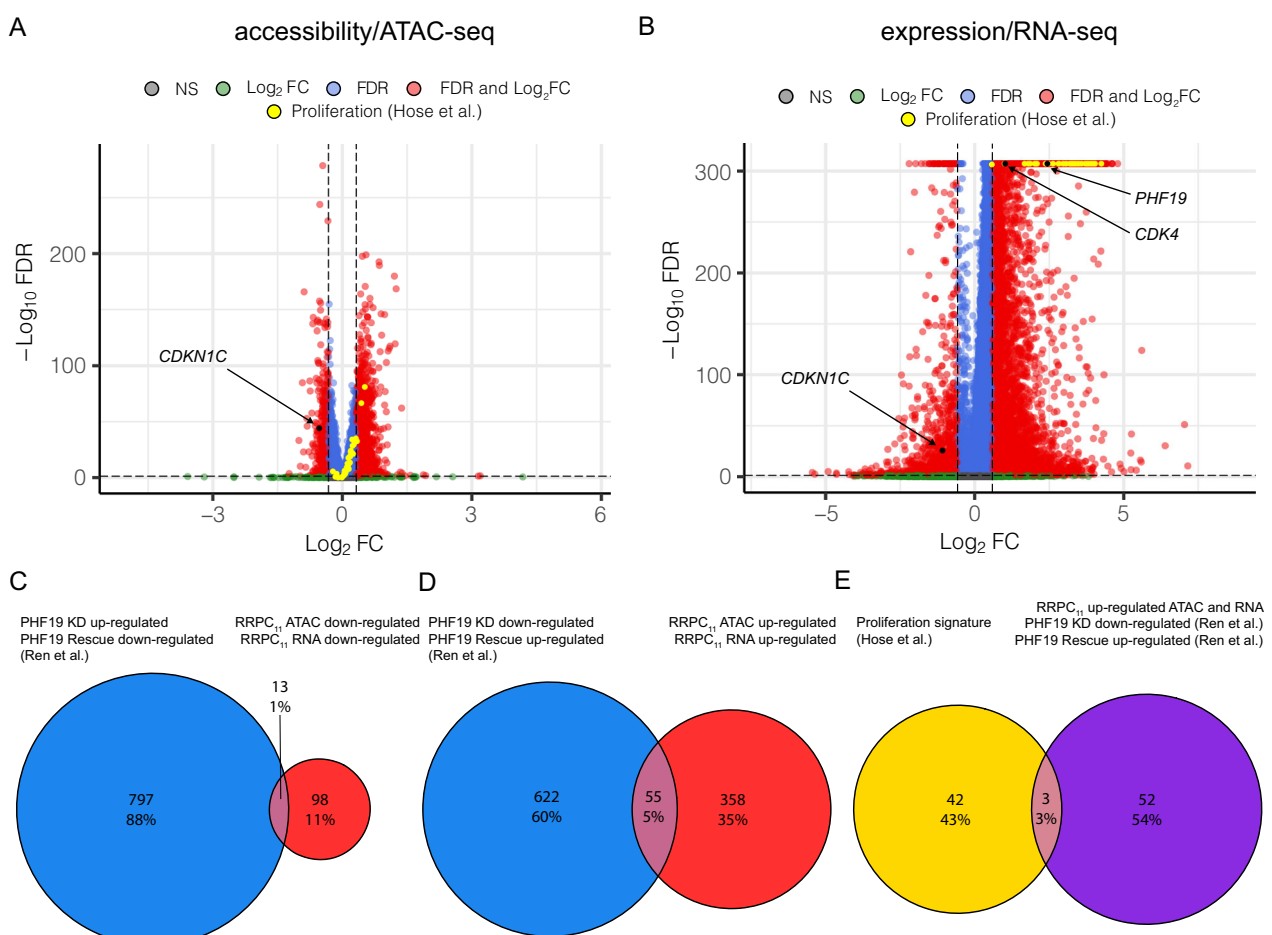

**Fig. 4 | Differential gene expression and chromatin accessibility in RRPC₁₁.**
**A** Gene activity differential chromatin accessibility for RRPC₁₁ with all proliferation genes annotated. **B** Differentially expressed genes for RRPC₁₁ with all proliferation genes annotated. **C** Intersection of PHF19 repressed genes from Ren et al. versus RRPC₁₁ ATAC and RNA downregulated genes. **D** Intersection of PHF19 up-regulated genes from Ren et al. versus RRPC₁₁ ATAC and RNA up-regulated genes. **E** Intersection of (**D**) with proliferation signature genes.

## PBX1 on 1q regulates expression of PHF19 in RRPCs

To further evaluate the connections between copy number variations and regulation of *PHF19* expression, transcription factors (TFs) were identified that were located on regions of the genome that were amplified (>3 copies) in patients with higher proportions of RRPCs. The only significant gain or amplification affecting all RRPC cluster proportions was Gain/Amp1q (Fig. 2D, Table 3). The TF2DNA[20] and hTFtarget[21] databases contain pairs of TFs and their potential targets. In total, 35 TFs were identified from TF2DNA and 57 TFs were identified from hTFtarget that target *PHF19*. Of these TFs, seven (*ATF3*, *KDM5B*, *PBX1*, *RBBP5*, *RFX5*, *USF1*, *ZNF648*) were located on 1q, of which *PBX1* (Log₂FC = 0.78, *P* < 0.001 and FDR < 0.001), *RFX5* (Log₂FC = 0.81, *P* < 0.001 and FDR < 0.001), and *RBBP5* (Log₂FC = 0.93, *P* < 0.001 and FDR < 0.001) were significantly up-regulated in RRPC₁₁ compared to all other clusters (Fig. 5A). *PBX1* silencing experiments that resulted in reduced live cell percentage and lower tumor weight also had ChIP-seq and RNA-seq data available[11]. This dataset was used to investigate PBX1 binding in myeloma cell lines. From the myeloma cell lines MM1S and U266, both of which have Gain/Amp1q, ChIP-seq peaks were identified in the promoter of *PHF19* that were in the top 4% of MM1S ChIP-seq peaks and the top 8% of U266 ChIP-seq peaks (Fig. 5B–D). These peaks were also located in super-enhancer region within the *PHF19* genomic loci (Fig. 5B, Supplementary Fig. 9B)[19]. From RNA-seq data of MM1S (Fig. 5E) and U266 (Fig. 5F), silencing *PBX1* via shRNA significantly decreased *PHF19* expression (*P* = 0.002, MM1S + U266 min-max scaled per cell line Scrbl vs P11). Similar patterns were also found in pre-B cell

acute lymphoblastic leukemia (ALL) cell line (697)[22]. In the pre-B ALL 697 cells, *PBX1* ChIP-seq peaks existed upstream of *PHF19* (Supplementary Fig. 10A–C) and cells treated with shRNA silencing *PBX1* had significantly decreased *PHF19* expression (Supplementary Fig. 10D). In our scRNA-seq, both *PHF19* (Log₂FC = 2.43 *P* < 0.001 and FDR < 0.001) and *PBX1* (Log₂FC = 0.78, *P* < 0.001 and FDR < 0.001) are upregulated in RRPC₁₁ (Fig. 5G).

When myeloma cell lines overexpressed *PBX1*, we saw the opposite effect of the knockdown of *PBX1* on *PHF19* expression (Fig. 5H–K). When *PBX1* was overexpressed in PCM6 (Fig. 5H), *PHF19* expression is significantly increased (Fig. 5I) and when *PBX1* was overexpressed in MM1S (Fig. 5J), *PHF19* expression was also increased (Fig. 5K). Overall, across both cell lines the PBX1 expression and PHF19 expression increased significantly (PBX1 *P* < 0.001, PHF19 *P* < 0.001, MM1S + PCM6 min-max scaled per cell line eVec vs oePBX1). The effect size from *PBX1* overexpression was much larger in the PCM6 cells (Log₂FC = 0.75) than the MM1S cells (Log₂FC = 0.06) corresponding inversely to their 1q CNV statuses. We see notable upregulation of cell cycle genes like *TOP2A* (Supplementary Fig. 11A, B, Supplementary Tables 11, 12, Supplementary Data 5, 6) and *MKI67* (Supplementary Fig. 11B, Supplementary Table 12) and down-regulation of immune related genes like *CXCL10* (Supplementary Fig. 11A, Supplementary Table 11). At the individual cell level, cells with Gain/Amp1q(*PBX1*) and Amp1q(*PBX1*) had increased expression of *PHF19* (Log₂FC(CPM) = 0.94, *P* < 0.001 and Log₂FC(CPM) = 1.51, *P* < 0.001, respectively). *PHF19* expression is detectable in 47% of RRPC₁₁ cells compared to only 9% of cells in other

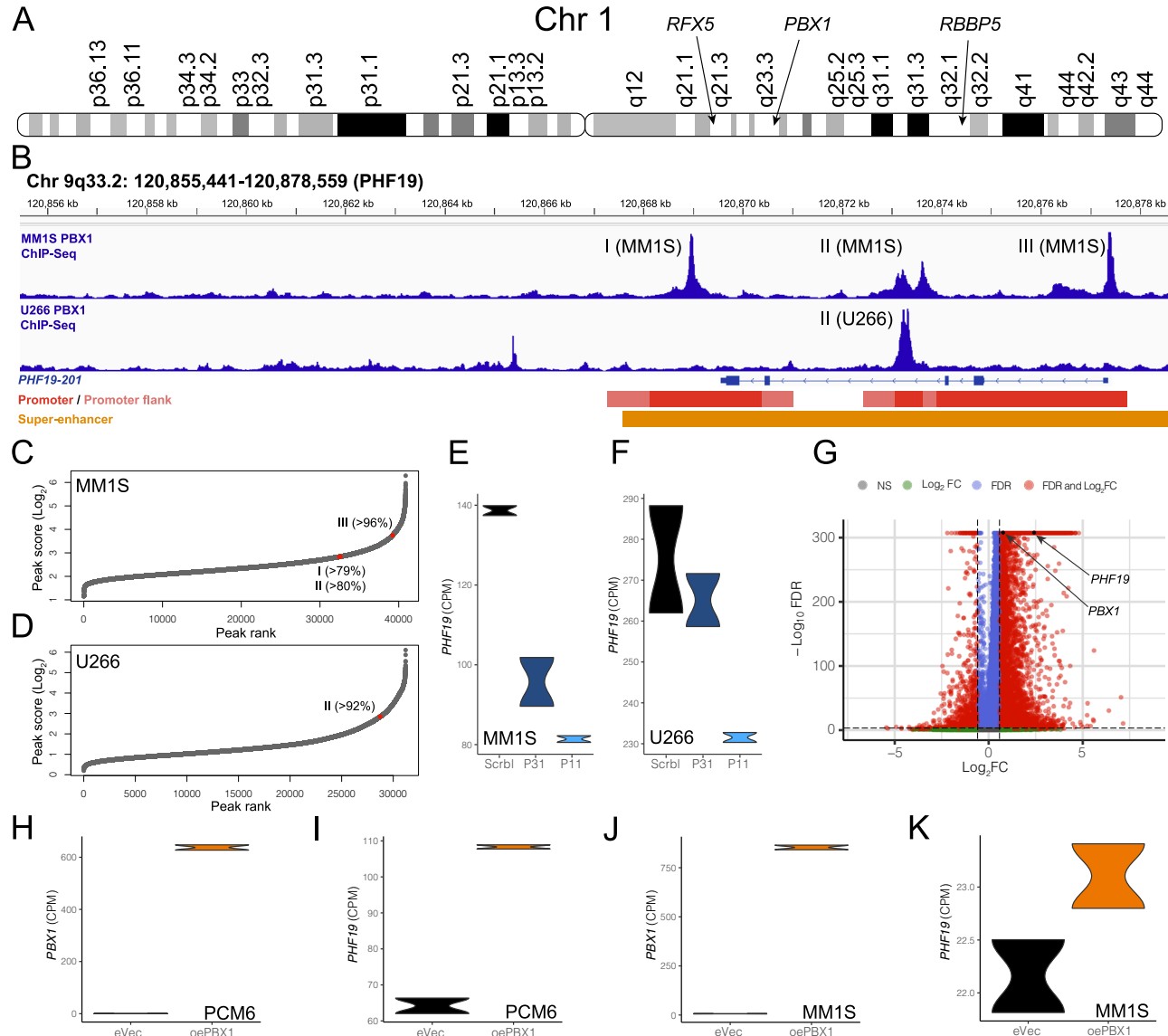

**Fig. 5 | Association of the chromosome 1q TFs with *PHF19* expression.**
**A** Location of TFs on chromosome 1q. **B** PBX1 ChIP-seq peaks in promoter and enhancer region of *PHF19* on chromosome 9. PBX1 ChIP-seq peak scores in MM1S (**C**) and U266 (**D**) myeloma cell lines with the percentile of each ChiP-seq peak marked. **E** Differences in *PHF19* expression in MM1S cells treated with PBX1 shRNAs. **F** Differences in *PHF19* expression in U266 cells treated with PBX1 shRNAs.

**G** Differential expression of *PBX1* and *PHF19* in RRPCs. **H** *PBX1* expression in PCM6 eVec cells compared to PCM6 oePBX1 cells. **I** *PHF19* expression in PCM6 eVec cells compared to PCM6 oePBX1 cells. **J** *PBX1* expression in MM1S eVec cells compared to MM1S oePBX1 cells. **K** *PHF19* expression in MM1S eVec cells compared to MM1S oePBX1 cells.

clusters and *PBX1* expression is detectable in 22% of RRPC$_{11}$ compared to only 10% of cells in other clusters, further strengthening their biological link. When we compared the correlation between *PBX1* and *PHF19* expression in the cohort of NDMM patients from the MMRF CoMMpass study, we again identified a significant correlation (PCC = 0.21, *P* < 0.001, >83.30% *PBX1* PCCs).

**Analysis of patient subclones indicates that those with Amp1q have increased expression of *PHF19***
To determine if we could identify the same co-expression in patient samples, we identified patients with subclones of Amp1q to determine if that subclone also had increased expression of *PHF19*. In the first patient (Fig. 6A–G), five clusters were identified from RNA (Fig. 6A), ATAC (Fig. 6B), and WNN integration of RNA and ATAC (Fig. 6C). Based on WGS, this patient had Amp1q and HRD (Fig. 6D), and major subclones could be distinguished based on inferred copy number of the

scRNA-seq data (Fig. 6E) with 3/5 subclones representing the majority of the cells. Clones 3 and 4 both had Amp1q in contrast to clones 1, 2, and 5 (Fig. 6E). *RBBP5*, *PBX1*, and *PHF19* were all significantly increased in expression (Fig. 6F) and had increased number of cells in G2M and S phases of cell cycle (Fig. 6G) in the clones with Gain/Amp1q. This may indicate that subclonal differences of Amp1q could affect *PHF19* expression through increased expression of the key TFs.

A second patient also contained subclonal heterogeneity represented by four clones that could be distinguished by clustering of RNA (Fig. 6H), ATAC (Fig. 6I), and WNN integration of RNA and ATAC (Fig. 6J). Based on WGS, this patient had Amp1q, and was hyperdiploid (Fig. 6K). Clone 4 had Amp1q (Fig. 6L), had significantly increased expression of *RBBP5*, *PBX1*, and *PHF19* compared to clones 1, 2, and 3 (Fig. 6M), and had more cells in G2M and S phase of cell cycle (Fig. 6N). These examples demonstrate that even at the subclonal level there is likely regulation of *PHF19* by TFs on chromosome 1q, leading to high-risk disease.

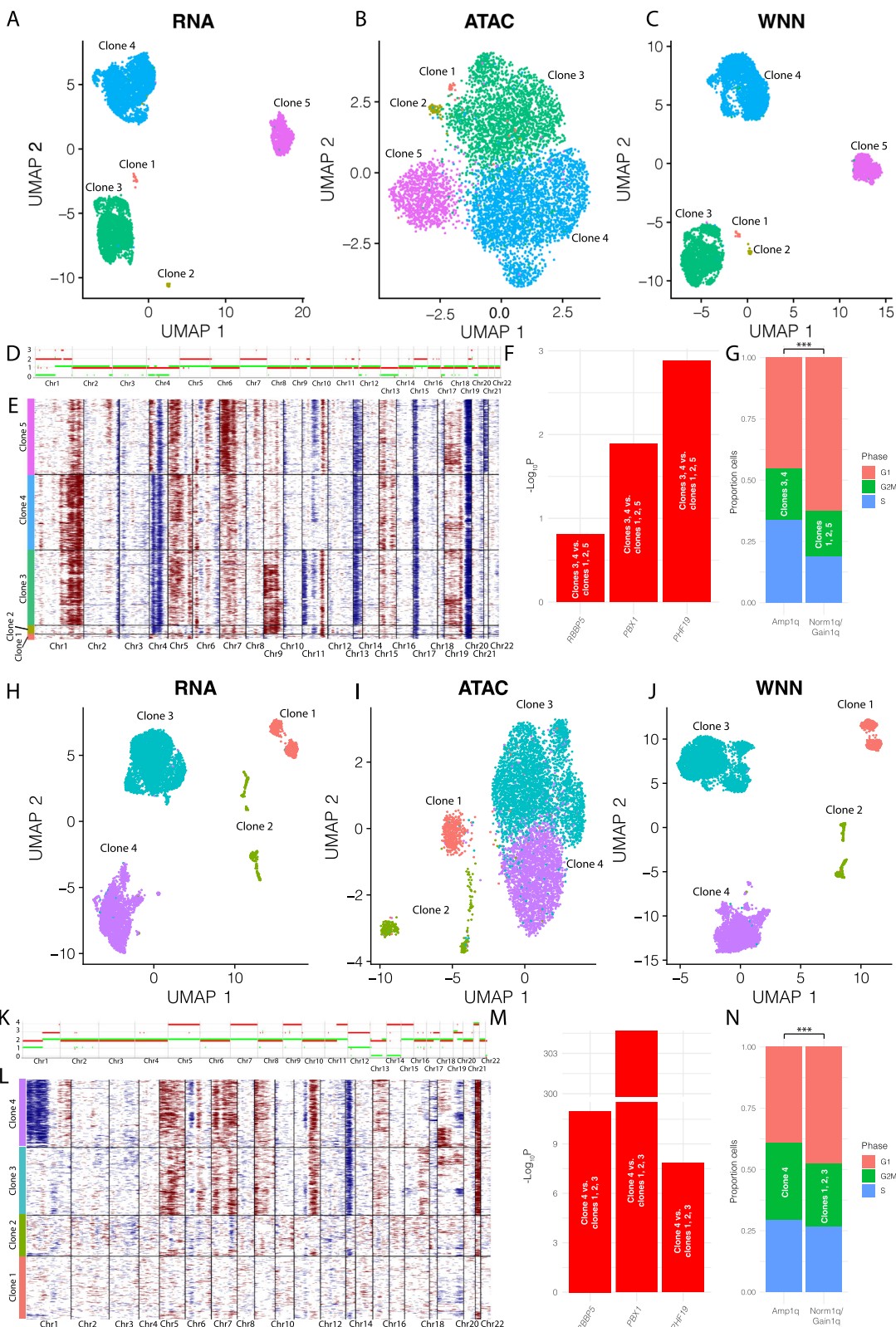

**Fig. 6 | Subclonal differences in Gain/Amp1q.** Patient sample 0661-533 clustering based on RNA (**A**), ATAC (**B**), WNN integrated RNA and ATAC (**C**). **D** WGS based CNV profile for 0661-533. **E** CNVs inferred from scRNA-seq data for 0661-533. **F** Increased expression of *PBX1*, *RBBP5*, and *PHF19* in Amp1q clones from 0661-533 (One-sided t-test: *RBBP5* P = 1.54E−2, *PBX1* P = 1.27E−2, *PHF19* P = 3.55E−3). **G** Increase in G2M and S phases of cell cycle for cells in Amp1q clones from 0661-533 (Fisher's exact P = 8.47E−26). Patient sample 0661-1043 clustering based on RNA (**H**), ATAC (**I**),

WNN integrated RNA and ATAC (**J**). **K** WGS based CNV profile. **L** CNVs inferred from scRNA-seq data. **M** Increased expression of *PBX1*, *RBBP5*, and *PHF19* in Amp1q clones (One-sided t-test: *RBBP5* P = 2.44E−11, *PBX1* P = 9.38E−170, *PHF19* P = 2.24E−6). **N** Increase in G2M and S phases of cell cycle for cells in Amp1q clones from 0661-1043 (Fisher's exact P = 3.69E−12). Fisher's exact test significance levels are: P < 0.1: (·), P < 0.05: *, P < 0.01 (**), P < 0.001 (***).

## Discussion

We report on a large single-cell multiomic study from patients across different stages of disease progression. Based on these data we have identified multiple proliferative clusters that are associated with specific genomic events and stages of myeloma progression. Notably, four such clusters denoted (RRPCs) significantly increase in proportion from SMM to RRMM and a subset of three (RRPC$_{11,20,22}$) are associated with Amp1q and express higher levels of *PHF19*. The largest subset of RRPCs, RRPC$_{11}$, are enriched predominantly in RRMM patients. RRPC$_{11}$ have increased expression of *PHF19*, a prognostic marker for myeloma, resulting in reduce chromatin accessibility and gene expression of *CDKN1C*. Not surprisingly these same cells exhibit a proliferation signature suggesting a regulatory role of *PHF19* in RRMM patients that results in a more proliferative state. It is intriguing that these high-risk RRPCs only make up <17% of a RRMM patient's myeloma cells. However, this is not to say that they do not have an outsized effect on patient survival or interact with other non-RRPCs to promote disease. In other cancers, intra-tumor heterogeneity leads to interactions between subclones, through cell signaling, leading to diffuse infiltration and complicating disease management[23]. Positive interactions from one subclone can lead to the population as a whole benefiting, resulting in commensalism[24]. Examples of such benefits could be secreted factors, weakening of the immune system, or stimulation of the bone marrow to provide a protective niche[23–26]. Determining the function and interactions of small high-risk clones with both other myeloma cells and the microenvironment will increase our understanding of resistance and relapsed disease.

Besides the clear association of RRPCs with myeloma progression, an additive association with *TP53* mutations and 1q CNVs was identified in these RRPCs. It is worth noting that we could not call mutations at the single cell level accurately using this type of single cell data. For these reasons we use the patient level mutational data from paired WGS to make inferences about the clusters. Once better methods become available, we will call mutations at the single cell level using either RNA or ATAC reads. Unlike mutations, CNVs can be called at the single cell level and we found a clear association of between 1q and *PHF19* using both patient level 1q alterations from WGS and single cell level 1q alterations from scRNA-seq. Because *PHF19* is not located on 1q, we evaluated TFs that may regulate *PHF19* located on 1q. The TFs *PBX1*, *RFX5*, and *RBBP5* were identified on 1q and may regulate *PHF19* based on KD experiments. RRPCs have consistent upregulation of these TFs and *PHF19* can also be identified in subclones containing 1q alterations. Taking all of this into consideration this data may indicate an association between *PHF19* and 1q via TF regulation creating unique molecular subtypes of myeloma cells.

Molecular subtypes of myeloma have been identified using analysis of gene expression data including a proliferation (PR) group with worse overall survival and progression free survival than other groups[2]. At the single cell level, proliferative subpopulations of abnormal plasma cells have been identified in myeloma patients showing that the proliferation signature is not uniformly distributed across plasma cells from a myeloma patient[27] and that they are more prevalent in relapsed patients[28]. Furthermore, proliferation was found to be the central prognostic factor for myeloma patients and Gain/Amp1q is highly correlated with higher proliferation index[11].

Given the high correlation between Gain/Amp1q and the proliferation signature, it is not surprising that Gain/Amp1q is a high-risk cytogenetic event in myeloma. Gain/Amp1q is the most important single event that confers higher risk of relapse[29]. Our results again highlight that the highest risk subclones with a proliferation signature were highly enriched for Gain/Amp1q. Further study on 1q is needed due to 1q alterations being clearly associated with worse progression, having an incompletely understood mechanism, and the risk of relapse greatly increases in patients with both a 1q alternation and TP53 mutations[30]. It is of the utmost importance to study mechanistically how 1q is conferring risk and also why there is synergy between 1q alterations, TP53 mutations, and proliferative cellular phenotypes. One such mechanism that has been proposed is the upregulation of the TF *PBX1* on 1q23 which causes upregulation of *FOXM1*[11]. Interestingly, we find there is also evidence that *PHF19* is simultaneously targeted by *PBX1* in Gain/Amp1q patients. Considering recent studies have also determined that *PHF19* expression is one of the most important single prognostic factors in myeloma[13], regulation of *PHF19* by 1q alterations would add some context to Gain/Amp1q mechanistically.

Further complicating the relationships between high-risk cytogentic events and progression, is clonal heterogeneity. Based on combined B-cell receptor V(D)J and RNA sequencing, multiple subclones were been identified form NDMM and RRMM patient samples with some convergent proliferative phenotypes associated with progression[31]. Frequently subclones with Gain/Amp1q tended to have a survival advantage during treatment and leading to an expansion of Gain/Amp1q subclones[32]. These survival advantages of specific subclones during treatment and relapse, also have an epigenetic regulatory component conferring resistance to therapy[33]. These recent results further support our findings linking high-risk copy number changes to epigenetic regulation of proliferation pathways.

In summary we have generated one of the largest single cell multiomic datasets for myeloma. From these data we have identified distinct proliferation states that are associated with high-risk cytogenetic events, identified three new TFs to study as they relate to 1q regulation of cell cycle, and discovered a strong link between Amp1q, *PHF19* expression, and proliferation. Specifically, *PHF19* likely epigenetically regulates cell cycle in a subset of cells found primarily in RRMM patients. Gain and especially amplification of PBX1 on 1q may contribute to this high *PHF19* expression phenotype found in these cells.

## Methods

### Sample collection and sequencing

Bone marrow aspirates from 10 smoldering multiple myeloma (SMM), 22 newly diagnosed multiple myeloma (NDMM), and 17 relapsed or refractory multiple myeloma (RRMM) patients were collected through the Indiana Myeloma Registry with informed consent and institutional approval. All sample collection in this study was reviewed by the Indiana University Human Research Protection Program Institutional Review Board and all participants consented in writing to be in this study. Samples were collected under approval 1804208190 (Indiana Myeloma Registry) and this specific study with deidentified samples was deemed exempt by the IUSCCC scientific review committee as the study was retrospective. These samples underwent CD138+ magnetic-activated cell sorting resulting in a CD138+ fraction with average purity $84.0 \pm 9.6\%$ and were viably frozen.

### Single-cell multiomic sequencing

Single-cell multiome analysis was conducted using a 10X Chromium system (10X Genomics, Inc). Cryopreserved cells were thawed and washed as previously described[34]. The cell suspensions were processed into RNA and ATAC libraries using the manufacturers standard protocol. The final single cell suspension was washed three times with PBS plus 0.04% BSA. If cell viability was less than 70%, a dead cell depletion procedure was applied using the MACS Dead Cell Removal Kit (Miltenyi). Each clean single cell suspension was counted for cell number and cell viability. Cells were lysed using digitonin lysis buffer, and single nuclei prepared based on the protocol of Nuclei Isolation from Mouse Brain Tissue for Single Cell ATAC Sequencing, CG000212 Rev B (10X Genomics, Inc). A final nuclei concentration of 3000/μL or higher was used for targeted cell recovery of 10,000 nuclei. Following the Chromium_NextGEM_Multiome_ATAC_GEX_User_Guide, CG000338_RevB (10X Genomics, Inc), tagmentation of nuclei preparation was performed. Thereafter, briefly, along with the single cell multiome gel beads and partition oil, the single nuclei master mixture containing tagmented

single nuclei suspension was transferred onto a Next GEM Chip J in separate wells, one sample per well, and the chip loaded to the Chromium Controller for GEM generation and barcoding, followed by pre-amplification PCR, ATAC library preparation, cDNA synthesis and cDNA library preparation. At each step, the quality of cDNA, ATAC library and cDNA library were examined by Bioanalyzer and Qubit. The resulting ATAC and cDNA libraries were sequenced separately, cDNA library for 28 bp and 91 bp paired-end and ATAC library for 50 bp paired-end sequencing on a NovaSeq 6000 (Illumina).

### Bulk DNA sequencing

CD138+ cells for 44 samples also underwent bulk whole genome sequencing (WGS) and targeted panel sequencing[35] to identify copy number alterations, translocations, and single nucleotide variations. For WGS, genomic DNA from tumor and non-tumor (saliva or peripheral blood sample from the same patient) samples were prepared using the DNA PCR-free Library Prep Tagmentation Kit (Illumina). Libraries were pooled and sequenced in 150 bp paired-end read format on a NovaSeq 6000 (Illumina) to a mean depth of 73× for tumor samples and 27× for matched control samples.

WGS samples were pre-processed using the Myeloma Genome Project 1000 (MGP1000) pipeline (https://github.com/pblaney/mgp1000). Coverage metrics and GC bias metrics were calculated for each BAM file. For each matched normal/tumor pair, genetic concordance and contaminations were estimated using Conpair (v0.2)[36] prior to variant analysis. Strelka (v2.9.2)[37] was used for variant calling and single nucleotide polymorphisms (SNPs) were filtered using fpfilter (https://github.com/ckandoth/variant-filter) to a 5% VAF cut-off[38]. Indels were filtered using a 10% VAF cut-off. Variants were annotated using Variant Effect Predictor (v101)[39]. Structural variants were calculated using Manta (v1.6.0)[40]. Copy number variations were analyzed using ASCAT-NGS[41] and were defined as follows: 0 copies (deep deletion), 1 copy (deletion), 2 (normal), 3 (gain), and 4+ (amplification).

### Cleaning and Preprocessing of CD138+ scRNA-seq

The single-cell multiomics reads for the CD138+ fraction were aligned and quantified using the cellranger-arc (v1.0.1). These aligned and quantified reads were used as input to a custom-built Seurat (v4.3.0) pipeline for single-cell multiomics that performed quality control (QC), multi-dataset integration, and non-plasma cell removal. For the integration experiment, high quality cells were identified through the following QC process. For each sample, the number of unique features and percentage of mitochondrial RNAs were plotted as violin plots. For each of the 49 samples these plots were evaluated to identify cutoffs based on the distribution of the data. A max number of unique RNAs, min number of unique RNAs, and a max percentage of mitochondrial RNAs were all set according to these initial violin plots. In a similar fashion, cells were removed from the multiomic integration experiments using the ATAC data as well. Violin plots of the number of unique ATAC features, nucleosome signal, and transcription start site (TSS) enrichment were plotted. Based on the distribution of each of these features a max and min cutoff value was set to remove outliers and low quality cells. A max number of unique ATAC features, min number of unique ATAC features, max nucleosome signal, min nucleosome signal, max TSS enrichment, and min TSS enrichment were all set according to these initial violin plots. For each dataset, QC was performed including removal of: low expression cells, cells with low number of features, cells with high number of features, and cells with high percentage mitochondrial RNA as outlined in Seurat documentation and online resources[42,43].

### Integration of scRNA-seq across samples and non-plasma cell removal

An iterative process was used to remove non-plasma cells from the dataset. In the first iteration, all 49 samples were integrated together based on their RNA profiles using Seurat (v4.3.0). The expression of *SDC1* (CD138) was summarized for each cluster using percentiles such that the 10th, 25th, 50th, 75th, and 90th percentile cells were used to summarize the expression in a given cluster. Clusters with *SDC1* expression with 0 counts in the 90th percentile cell were removed as contaminant cells. After contaminant cell removal, a second iteration of integration was performed on the remaining cells. In this iteration, clusters were removed if *SDC1* expression was 0 counts in the 75th percentile cell of that cluster. The remaining clusters were used for the remainder of the analysis to study changes in myeloma cells across stages of progression. Samples were annotated based on the WGS data for common CNVs and translocations, which were confirmed by expression of translocation markers (*NSD2, FGFR3, CCND1, CCND3, MAF, MAFB, ITGB7, CCND2, MYC*) in the single-cell data.

### Identification of high-risk myeloma cells from scRNA-seq

For each of the identified plasma cell clusters derived from the CD138+ samples, the proportion of that cluster was calculated for each patient such that each patient had a total proportion of 1.0 split between each individual cluster. For example, if a patient had 1000 cells where 500 cells were in cluster 1, 200 cells were in cluster 2, and 300 cells were in cluster 3, the cluster proportions for that patient would be 0.5 cluster 1, 0.2 cluster 2, and 0.3 cluster 3. This was repeated for each patient using the clusters from the integrated Seurat object. The cluster proportions across all of the patients were compared to high-risk copy number alterations, disease status, and mutation events to identify clusters associated with these variables.

We performed comprehensive hypothesis testing to evaluate whether clusters were enriched in specific genomic, cytogenetic, and clinical patient groups. Each pairwise comparison was evaluated for the proportion of clusters 1-25 and patient covariates: RRMM vs SMM, NDMM vs SMM, RRMM vs NDMM, age ≥65 vs age <65, t(4;14) vs not t(4;14), t(11;14) vs not t(11;14), t(6;14) vs not t(6;14), t(14;16) vs not t(14;16), t(14;20) vs not t(14;20), HRD vs not HRD, t(*MYC*) vs not t(*MYC*), *MYC* abnormality vs no *MYC* abnormality, Del(*CDKN2C*) vs Norm(*CDKN2C*), Del(*RPL5*) vs Norm(*RPL5*), Del(*TENT5C*) vs Norm(*TENT5C*), Del(*PRKN*) vs Norm(*PRKN*), Del(*DOCK5*) vs Norm(*DOCK5*), Del(*CDKN2A*) vs Norm(*CDKN2A*), Del(*BIRC3*) vs Norm(*BIRC3*), Del(*ATM*) vs Norm(*ATM*), Del(*CDKN1B*) vs Norm(*CDKN1B*), Del(*BRCA2*) vs Norm(*BRCA2*), Del(*RB1*) vs Norm(*RB1*), Del(*DIS3*) vs Norm(*DIS3*), Del(*ABCD4*) vs Norm(*ABCD4*), Del(*TRAF3*) vs Norm(*TRAF3*), Del(*CYLD*) vs Norm(*CYLD*), Del(*WWOX*) vs Norm(*WWOX*), Del(*MAF*) vs Norm(*MAF*), Del(*TP53*) vs Norm(*TP53*), Gain/Amp(*CKS1B*) vs Norm(*CKS1B*), Amp(*CKS1B*) vs Norm(*CKS1B*), Gain/Amp(*TNFAIP8*) vs Norm(*TNFAIP8*), Amp(*TNFAIP8*) vs Norm(*TNFAIP8*), Gain/Amp(*TNXB*) vs Norm(*TNXB*), Amp(*TNXB*) vs Norm(*TNXB*), Gain/Amp(*ADCY2*) vs Norm(*ADCY2*), Amp(*ADCY2*) vs Norm(*ADCY2*), Gain/Amp(*RAPGEF5*) vs Norm(*RAPGEF5*), Amp(*RAPGEF5*) vs Norm(*RAPGEF5*), Gain/Amp(*KLF14*) vs Norm(*KLF14*), Amp(*KLF14*) vs Norm(*KLF14*), Gain/Amp(*MYC*) vs Norm(*MYC*), Amp(*MYC*) vs Norm(*MYC*), Gain/Amp(*CDKN2A*) vs Norm(*CDKN2A*), Amp(*CDKN2A*) vs Norm(*CDKN2A*), Gain/Amp(*RNF20*) vs Norm(*RNF20*), Amp(*RNF20*) vs Norm(*RNF20*), Gain/Amp(*TRAF2*) vs Norm(*TRAF2*), Amp(*TRAF2*) vs Norm(*TRAF2*), Gain/Amp(*RRAS2*) vs Norm(*RRAS2*), Amp(*RRAS2*) vs Norm(*RRAS2*), Gain/Amp(*CCND1*) vs Norm(*CCND1*), Amp(*CCND1*) vs Norm(*CCND1*), Gain/Amp(*BIRC*) vs Norm(*BIRC*), Amp(*BIRC*) vs Norm(*BIRC*), Gain/Amp(*ATM*) vs Norm(*ATM*), Amp(*ATM*) vs Norm(*ATM*), Gain/Amp(*CDKN1B*) vs Norm(*CDKN1B*), Amp(*CDKN1B*) vs Norm(*CDKN1B*), Gain/Amp(*DNMT3A*) vs Norm(*DNMT3A*), Amp(*DNMT3A*) vs Norm(*DNMT3A*), Gain/Amp(*CRBN*) vs Norm(*CRBN*), Amp(*CRBN*) vs Norm(*CRBN*), Gain/Amp(*FGFR3*) vs Norm(*FGFR3*), Amp(*FGFR3*) vs Norm(*FGFR3*), Gain/Amp(*ABCD4*) vs Norm(*ABCD4*), Amp(*ABCD4*) vs Norm(*ABCD4*), Gain/Amp(*TRAF3*) vs Norm(*TRAF3*), Amp(*TRAF3*) vs Norm(*TRAF3*), Gain/Amp(*WDR72*) vs Norm(*WDR72*), Amp(*WDR72*) vs Norm(*WDR72*), Gain/Amp(*BLM*) vs Norm(*BLM*), Amp(*BLM*) vs Norm(*BLM*), Gain/

Amp(*AKAP1*) vs Norm(*AKAP1*), Amp(*AKAP1*) vs Norm(*AKAP1*), Gain/Amp(*ZNF426*) vs Norm(*ZNF426*), Amp(*ZNF426*) vs Norm(*ZNF426*), Gain/Amp(*ZNF227*) vs Norm(*ZNF227*), Amp(*ZNF227*) vs Norm(*ZNF227*), Gain/Amp(*SON*) vs Norm(*SON*), Amp(*SON*) vs Norm(*SON*), Mut(*KRAS*) vs not Mut(*KRAS*), Mut(*NRAS*) vs not Mut(*NRAS*), Mut(*TENT5C*) vs not Mut(*TENT5C*), Mut(*DIS3*) vs not Mut(*DIS3*), Mut(*BRAF*) vs not Mut(*BRAF*), Mut(*TRAF3*) vs not Mut(*TRAF3*), Mut(*TP53*) vs not Mut(*TP53*), Mut(*CYLD*) vs not Mut(*CYLD*), Mut(*MAX*) vs not Mut(*MAX*), Mut(*UBR5*) vs not Mut(*UBR5*), Mut(*USP7*) vs not Mut(*USP7*), Mut(*IRF4*) vs not Mut(*IRF4*), Mut(*SP140*) vs not Mut(*SP140*), Mut(*PTPN11*) vs not Mut(*PTPN11*), Mut(*RB1*) vs not Mut(*RB1*), Mut(*NFKBIA*) vs not Mut(*NFKBIA*), Mut(*RASA2*) vs not Mut(*RASA2*), Mut(*TGDS*) vs not Mut(*TGDS*), Mut(*CDKN1B*) vs not Mut(*CDKN1B*), Mut(*DUSP2*) vs not Mut(*DUSP2*), Mut(*KLHL6*) vs not Mut(*KLHL6*), Mut(*EGR1*) vs not Mut(*EGR1*), Mut(*FGFR3*) vs not Mut(*FGFR3*), Mut(*ACTG1*) vs not Mut(*ACTG1*), Mut(*ZNF292*) vs not Mut(*ZNF292*), Mut(*HUWE1*) vs not Mut(*HUWE1*), Mut(*CCND1*) vs not Mut(*CCND1*), Mut(*SAMHD1*) vs not Mut(*SAMHD1*), Mut(*ABCF1*) vs not Mut(*ABCF1*), Mut(*CDKN2C*) vs not Mut(*CDKN2C*), Mut(*FUBP1*) vs not Mut(*FUBP1*), Mut(*PRDM1*) vs not Mut(*PRDM1*), Mut(*KMT2B*) vs not Mut(*KMT2B*), Mut(*ATM*) vs not Mut(*ATM*), Mut(*KMT2C*) vs not Mut(*KMT2C*), Mut(*CREBBP*) vs not Mut(*CREBBP*), Mut(*ARID1A*) vs not Mut(*ARID1A*), Mut(*ATRX*) vs not Mut(*ATRX*), Mut(*NF1*) vs not Mut(*NF1*), Mut(*TET2*) vs not Mut(*TET2*), Mut(*KDM5C*) vs not Mut(*KDM5C*), Mut(*ARID2*) vs not Mut(*ARID2*), Mut(*DNMT3A*) vs not Mut(*DNMT3A*), Mut(*KDM6A*) vs not Mut(*KDM6A*), and Mut(*MAFB*) vs not Mut(*MAFB*).

For each comparison (*i*) listed above, the mean proportion (*Prop*) of cluster *j* in across each patient was compared between the groups (*Group*) using a one-sided t-test. For each combination of comparison (*i*) and cluster (*j*) a *T*, *P*, -Log$_{10}$P, and Log$_2$FC, were calculated based on pairwise t-tests.

$$T_{ij}, P_{ij} = ttest(Prop_j, Group_i) \tag{1}$$

To visualize these results, the Log$_2$FC (first group over second group in each comparison e.g. Log$_2$(RRMM/SMM) for the RRMM vs SMM comparison) was used to color the dots in the dot plot. The -Log$_{10}$P was used to scale the size of each dot so that more significant cluster proportion-covariate comparison had larger dots. Only covariates with at least one significant *P* were retained in the dot plot. The FDR was also calculated for all of these comparisons. We considered a comparison significant if it's $P \leq 0.05$ and FDR $\leq 0.25$.

For high-risk clusters, DEG analysis was conducted on the RNA-seq and DAC analysis on the ATAC-seq. Firstly, the ATAC-seq data was aggregated to the gene level using the gene activity estimates provided by the Signac (v1.4.0) package. The DEG and DAC analyses were conducted by calculating the Log$_2$FC and Wilcoxon test *p*-values for each gene. The Benjamini-Hochberg correction was used for multiple testing correction of the *p*-values. For the DEG analysis, an absolute Log$_2$FC > 1.5 and FDR < 0.05 were used as cutoffs. For the DAC analysis, an absolute Log$_2$FC > 1.25 and FDR < 0.05 were used as cutoffs.

Besides the DEGs and DACs identified from our own data, DEGs were also used from previous *PHF19* knock down (KD) experiments[14] from the differential expression table for the *PHF19* KD and *PHF19* rescue experiments including the Log$_2$FC, *p*-values, and adjusted *p*-values with a fold change cutoff (FC = 2) and a *p*-value cutoff of 0.05. Besides the *PHF19* KD signature, previously published proliferation signatures[2,18] were also included for comparison purposes. The intersections of DEGs, DACs, *PHF19* KO DEGs, and other known myeloma signatures were used to filter the genes into high-risk subsets for further study.

To better understand the biology underlying different subsets of cells from the clustering results, we performed functional enrichment analysis for the RRPC clusters. The DEG markers for each of the RRPC clusters were used as input to the clusterProfiler (4.0.5) program to identify enriched Gene Ontology (GO) terms. The enrichGO function uses a hypergeometric test to identify GO gene sets that overlap significantly with a set of DEGs. These terms were then displayed as a dot plot to show the number of genes and the significance level of each term. The GeneRatio denotes the percentage of the DEGs that are also contained in a specific GO term. The count variable denotes the number of genes in the numerator of the GeneRatio, i.e. the number of DEGs also contained in the GO term. Aside from studying the functional enrichment of our DEGs with GO terms, it was also important to study the overlap of DEGs with other gene sets from previous studies. For these purposes, we utilized Fisher's exact test to calculate the P and used the Odds Ratio (OR) to study the effect size.

## Individual multiomic integration and inference of CNVs at the single-cell level

Aside from the genomic alterations that were measured from WGS, it was also important to evaluate the subclonal structure within each sample and to impute samples that were missing WGS information. Translocations were imputed in the five samples missing WGS information, 0661-1097, 0661-1146, 0661-1184, 0661-1274, and 0661-6 using the canonical marker expression for each translocation These translocation and canonical markers included: t(4;14) (NSD2), t(6;14) (CCND3), t(11;14) (CCND1), t(14;16) (MAF), t(14;20) (MAFB), and t(MYC) (MYC). However, the t(MYC) results were inconclusive so t(MYC) was not imputed from MYC expression. Therefore, samples missing WGS were not included in analyses including t(MYC). First the CPM of each canonical marker ($x_{trans}$ where $trans \in \{transPos, tansNeg\}$) was calculated for each patient with known translocation status. The mean expression of the canonical marker was calculated for the translocation positive samples ($\bar{x}_{transPos}$) and translocation negative samples ($\bar{x}_{transNeg}$). A cutoff was defined as follows:

$$cutoff = \frac{(2\bar{x}_{transNeg} + \bar{x}_{transPos})}{3} \tag{2}$$

Any sample CPM falling above this cutoff was considered to have the translocation.

Imputation of CNVs was a slightly more complex process that required using two types of data, canonical markers and inferred CNVs from inferCNV (v1.8.1). CNVs were imputed in the five samples missing WGS information, 0661-1097, 0661-1146, 0661-1184, 0661-1274, and 0661-6. First the CPM of each canonical marker ($x_{cnv}$ where $cnv \in \{0,1,2,3,\geq 4\}$) was calculated for each patient with known CNV status. The mean expression of the canonical marker was calculated for the each CNV status ($\bar{x}_{cnv}$). The samples requiring imputation were preliminarily assigned the status of the CNV with closest $\bar{x}_{cnv}$ to the canonical marker CPM of that sample. Next the inferred CNV results across all the cells in that sample were reviewed using the inferCNV (v1.8.1) results. If there was strong evidence for a CNV (i.e., an indisputable blue band for deletions or indisputable red band for gain/amplification from the inferred CNV result), the preliminary assignment was changed according to the inferred CNV results. HRD was also evaluated for each of these five samples. If multiple full chromosomes was visibly duplicated in the inferred CNV results, then that sample was considered HRD.

The subclonal structure may be reflected in both the RNA-seq, ATAC-seq, and integrated clusters. For these reasons, multiomic integration was performed for each sample using Seurat (v4.3.0) and Signac (v1.4.0). Specifically, for each sample, the ATAC-seq count matrices and RNA-seq count matrices were converted into Seurat objects where the same non-plasma cell remove process was performed as in the original 49-sample integration for each sample individually. The ATAC-seq and RNA-seq data for these plasma cell clusters were integrated using the weighted nearest neighbor (WNN) algorithm from the Seurat (v4.3.0) package. These WNN-clusters were used as

our clustering variable for single cell CNV inference. For this analysis, the package inferCNV (v1.8.1)[44] was used to identify CNVs at the single cell level. Based on the previous analysis of plasma cell clusters, the normal plasma cell cluster that was identified was used as the reference for inferCNV (v1.8.1). Single cell CNVs were also calculated using a reference free software package, CopyKAT (v1.0.8)[45], to get location specific CNVs instead of gene specific. From these analyses the CNVs that correspond to clustering in the RNA-seq and ATAC-seq data were evaluated.

### Identification of *PHF19* promoter binding transcription factors (TF) and TF ChIP-seq peaks

Based on the preliminary analyses, the relationship between *PHF19* and high-risk copy number alterations that may contain TFs were evaluated. The TF databases, hTFtarget[21] and TF2DNA[20] were used to identify TFs that regulate *PHF19*. The identified *PHF19*-regulating TFs were cross-referenced against high-risk CNVs in myeloma. Paired ChIP-seq and RNA-seq in myeloma[11] were used to evaluate whether there were ChIP peaks upstream of *PHF19*. Both the existence of peaks within 5000 bp of the 5′ UTR as well as their relative strength in comparison to the other peaks across the genome were used to evaluate whether they likely indicate TF regulation of *PHF19*. Specifically, all of the identified peaks in the ChIP-seq experiments were ranked and the percentile of the peaks upstream of *PHF19* was used to discern how likely the TF was regulating *PHF19* opposed to the other peaks.

### Functional evaluation of PBX1 TF activity on PHF19 using RNA-seq

Previous studies already conducted knockdown experiments of *PBX1* using two shRNAs in the cell lines MM1S and U266 where global expression was measured using RNA-seq. Processed data were downloaded from GSE165060. For complete methods please see corresponding publication. We used a one-sided t-test in each cell line to test whether *PHF19* expression was reduced in each *PBX1* shRNA group. Preprocessed ChIP-seq and RNA-seq data was also downloaded from GSE138031. Specifically, MACS peak call BED and visualized peaks TDF files were downloaded for both PBX1 ChIP-seq and E2A/PBX1 ChIP-seq. The RNA-seq RPKM values were also downloaded for Scramble, *E2A*, and *PBX1* shRNA ($N = 2$ per group). Based on these data, the relative rank of peaks upstream of PHF19 were evaluated and the expression of PHF19 was compared between shRNA treated groups using a one-sided t-tests.

To further strengthen these relationships, we also performed a *PBX1* overexpression experiment in two myeloma cell lines, one with gain/amplification 1q (MM1S, Walker lab) and one without gain/amplification 1q (PCM6, Riken Catalogue #: RCB1460). HEK293T was obtained from American Type Culture Collection (ATCC). HEK293T cells were cultured in DMEM containing 10% FBS and 1% Penicillin-Streptomycin. MM1.S cells were cultured in RPMI containing 10% FBS and 1% Penicillin-Streptomycin. PCM6 cells were cultured in McCoy's 5A modified medium containing 20% FBS, 1% Penicillin-Streptomycin, and 3 ng/mL rIL-6. All cells were maintained at a temperature of 37 °C and $CO_2$ concentration of 5%. Plasmids used in this study were pLenti-C-Myc-DDK-P2A-puro (Origene PS100092), PBX1 (NM_002585) Human Tagged Lenti ORF Clone pLenti-C-Myc-DDK-P2A-puro (Origene RC210944L3), VSV-G, and PSPAX2.

To produce lentivirus, $1 \times 10^6$ HEK 293 T cells each were plated in two 10 cm$^2$ cell culture dishes with antibiotic free media. After overnight incubation at 37 °C and $CO_2$ concentration of 5%, 293T cells were transfected using Lipofectamine 3000. For one plate pLenti-C-Myc-DDK-P2A-puro (Origene PS100092) 1.7 µg, PSPAX2 2.5 µg, and VSV-G 764 ng were combined with 5 µL P3000 reagent and 5 µL L3000 reagent in 250 µL Opti-MEM. The mixture was incubated at room temperature for 15 minutes and added to the cells. For the other 10 cm$^2$ plate, Human Tagged Lenti ORF Clone pLenti-C-Myc-DDK-P2A-puro (Origene RC210944L3) 1.87 µg, 2.4 µg PSPAX2, and 730 ng VSV-G

plasmid were combined with 5 µL P3000 reagent and 5 µL L3000 reagent in 250 µL Opti-MEM. The mixture was incubated at room temperature for 15 min and added to the cells. After 48 h, lentiviral supernatant was collected and filtered with 0.45 µm syringe filter to remove cells and debris. The lentiviral supernatant was aliquoted and stored at −80 °C.

For lentiviral transduction, MM1.S and PCM6 cells were seeded ($1 \times 10^6$ per well in 12-well plates) in the appropriate media supplemented with 2 µg/mL polybrene. Lentivirus supernatant aliquots were thawed on ice for transduction. The amount of lentiviral supernatant was titrated for each cell line. Cells were spinfected at $400 \times g$ at room temperature for 1 h. Cells were then incubated at 37 °C and 5% $CO_2$ for 24 h. After the 24 h incubation, the transduced cells were selected with puromycin. MM1.S cells and PCM6 cells were transduced with either pLenti-C-Myc-DDK-P2A-puro (Origene PS100092) as an empty vector control or, PBX1 (NM_002585) Human Tagged Lenti ORF Clone pLenti-C-Myc-DDK-P2A-puro (Origene RC210944L3). Transduced MM1.S and PCM6 cells were selected in the recommended media supplemented with 0.5 µg/mL puromycin. Wild type MM1.S and PCM6 cells that were not transduced were treated with puromycin as a kill control. The lowest amount of virus that produced puromycin resistant cells was selected for transductions.

Resistant cells from both *PBX1* and empty vector transduced cells were grown and subsequently RNA was extracted in duplicate using the RNeasy Mini kit (Qiagen). 100 nanograms of total RNA was used for library preparation utilizing the Illumina Stranded mRNA Prep kit. 150 bp paired-end reads were generated on an Illumina NovaSeq 6000 sequencer with a target depth of 50 M reads. Reads were aligned to the GRCh38 primary assembly with GENCODE comprehensive gene annotation for reference chromosomes (release 45) using STAR (v2.7.11a). The aligned reads were then quantified to their respective genes using featureCounts (v2.0.6).

For both myeloma cell lines (MM1S and PCM6) quality control was performed to ensure high quality cells for our analysis. Targeted sequencing was performed and mutations and copy number changes were verified against data held elsewhere (https://www.keatslab.org/myeloma-cell-lines/hmcl-characteristics). PCR based mycoplasma testing was performed with the Promokine PCR Mycoplasma test kit (catalog number PK-CA91-1024) and neither cell line tested positive against the included positive control. After these cell lines were grown and processed with either *PBX1* overexpression (oePBX1) or empty vector (eVec) control, the expression of *PHF19* was compared using a one-sided t-test. The global gene expression changes were also analyzed with EdgeR (3.34.1)[46] where a gene was considered a DEG if |Log$_2$FC|>1 and FDR of <1E−5.

### Statistics and reproducibility

We determined that sample sizes of 10 in each diagnosis group (SMM, NDMM, RRMM), should achieve significance based on a power analysis (One-sided unpaired t-test, Cohen's D > 1.35, Power = 0.80). Sample sizes beyond 10 per group were dependent on sample availability in the Indiana Myeloma Registry. No data were excluded from the analyses unless it did not pass specified QC. The experiments in this study did not require randomization because there were no experimental treatment arms in this study. The researchers were not blinded to the patient and cellular metadata in this study.

### Reporting summary

Further information on research design is available in the Nature Portfolio Reporting Summary linked to this article.

### Data availability

The single cell multiomic data, WGS data, and accompanying metadata have been deposited in the dbGAP database under accession code phs003220 (http://www.ncbi.nlm.nih.gov/projects/gap/cgi-bin/study.cgi?study_id=phs003220.v2.p1). The single cell multiomic data and

WGS data is available under controlled access and can be obtained by submitting a project request through dbGaP. Projects will be approved for data access if they are for research purposes and if the investigator is sponsored at an appropriate research institution. A response should be received after the request within two weeks and the data will be available for 12 months with optional renewal after approval. The processed single cell multiomic data has been deposited in the Synapse database under accession code syn52295155 (www.synapse.org/#!Synapse:syn52295155). The newly generated RNA-seq data have been deposited in the GEO database under accession code GSE254307. The single cell multiomic, WGS, and RNA-seq datasets that were generated in this study were processed with GRCh38. The single cell multiomic data used the refdata-cellranger-arc-GRCh38-2020-A version while the RNA-seq data used the GRCh38 primary assembly with GENCODE comprehensive gene annotation for reference chromosomes (release 45). Publicly available RNA-seq and ChiP-seq datasets were retrieved from GEO (GSE165060[11] and GSE138031[22]). The myeloma RNA-seq data used in the study was from the MMRF CoMMpass study IA18 (http://research.themmrf.org). The TF databases were downloaded from the following sources htTFtarget[21] (http://bioinfo.life.hust.edu.cn/hTFtarget#!/download) and TF2DNA[20] (https://www.fiserlab.org/tf2dna_db/downloads.html). The remaining data are available within the Article, Supplementary Information or Source Data files (https://doi.org/10.6084/m9.figshare.25563525).

## Code availability

Scripts used for the analysis in this manuscript can be found at the following GitHub site: https://github.com/tsteelejohnson91/MM_CD138pos_scripts.

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

## Acknowledgements

The Indiana Myeloma Registry is funded in part by support from the Indiana University Precision Health Initiative, Miles for Myeloma, the Harry and Edith Gladstein Chair, and the Omar Barham Fighting Cancer Fund. Computational infrastructure at Indiana University (Indianapolis, IN) was funded in part by Lilly Endowment Inc. through the Indiana University Pervasive Technology Institute. The authors acknowledge the efforts of the Multiple Myeloma Research Foundation research consortium to fund and provide the fundamental CoMMpass resource for our study. We thank the Center for Medical Genomics at the Indiana University School of Medicine, a core supported by the NCI Cancer Center P30 support grant CA082709, for their expertise in carrying out single-cell genomic studies, especially Xiaoling Xuei and Yunlong Liu. R.A., B.A.W., and M.A.Z. received research support from Genentech to carry out this study. B.A.W. is partly funded by the Daniel and Lori Efroymson Chair. T.S.J. received a research fellowship grant from the Multiple Myeloma Research Foundation, National Cancer Institute grant R21CA264339, International Myeloma Society travel funds, and is partly funded by the Agnes Beaudry Investigator in Myeloma Research fund. E.L. received a research fellowship grant from the Multiple Myeloma Research Foundation.

## Author contributions

T.S.J., B.W., M.A. and R.A. designed the research. T.S.J., W.W., P.S., E.L., V.C., C.D., M.N., K.H., A.S., M.A. and R.A. performed the research. B.W. and M.A. collected the data. T.S.J., B.W., P.S., E.L., N.B., S.R., P.B., G.M., V.C., C.D., M.N., K.H., A.S., M.A. and R.A. analyzed and interpreted the data. T.S.J., P.S., and E.L. performed statistical analysis. T.S.J. and B.W. wrote the manuscript.

## Competing interests

V.C., C.D., and M.N. are Genetech/Roche employees and hold Roche stock options. The remaining authors declare no competing interests.
