## [Peer Review File · Nature Communications]

1q amplification and PHF19 expressing high-risk cells are associated with relapsed/refractory multiple myelomaReviewers' Comments:

Reviewer #1:

Remarks to the Author:

Understanding the pathogenesis and biology behind 1q amplification and its impact on myeloma cells is one of the most important questions in MM biology. This manuscript provides an exceptionally rich resource with WGS, RNAseq and single cell studies (ATAC and scRNAseq) to provide a comprehensive genomic dataset. The single cells studies are particularly important because they are novel, but also because they allow the dissection of subclonal heterogeneity that has been obscured in prior studies. Needless to say, synthesizing all of this diverse data into a comprehensible analysis is a huge challenge, which the authors have met admirably.

Most scRNA in MM has been on unselected BM cells, this is the first to focus on the heterogeneity within the tumor cell compartment. Important the cluster 11 (largest one associated with progression) increased from 1% > 3% > 8% with SMM > MM > RRMM. This is reminiscence of the plasma cell labelling index which (using much older technology) identified a similar fraction of proliferative cells. The big advantage now is that the genomic information for these cells is also captured, and can be analyzed.

The focus on PHF19 is appropriate, given it's outside impact on prognosis. The functional studies show clear PHF19-dependent cell proliferation. Importantly they show that PHF19 is a PBX1 target gene providing one plausible pathogenic scenario in which 1q amplification leads to increased pBX1 expression which drives PHF19 and FOXM1 expression.

These studies (and those in reference 11) show MM growth is dependent on PBX1 and PHF19, the next step for future studies is to show that over-expression of these factors can accelerate MM growth, the requirement to consider these driver genes.

Fig S3. Not clear what numbers in parentheses represent.

Not clear how del1p, gain1q, del13, del17p defined. Was CN of CDKN2C, CKS1B, RB1 and TP53 used? If so, list in the figures the gene name of the CN plotted (as they do in other figures). Provide the criteria they used to define these CAN.

Fig 3A is hard to interpret. Presumably 1-25 refer to the clusters, and each vertical line is an individual patient. It is not clear if what is plotted is the number of cells which express the given marker, or the average expression of the marker among the patients cells in the cluster. It would be helpful to order the genes based on upregulation and downregulation in the high risk clusters. It is not clear to me that any genes are downregulated. Perhaps a different normalization would help with this.

Fig 4D. The co-expression of PHF19 and MKI67 is not convincing. It looks like some cells are predominantly red, some predominantly green, and only 1 yellow (expressing both). I think an X-Y plot would be more convincing. I would expect that some patients may have biallelic deletions of CDKN2C, or RB1 with high MKI67 without elevated PHF19, which might become more evident.

Reviewer #2:

Remarks to the Author:

1. Summary from reviewer:

- a. This article investigates data from of 49 scMultiomics (10 SMOL, 22 NDMM and 17 RMM), as well as 44 WGS samples.
- b. Initially, authors characterized each sample in terms of mutations, copy number, cytogenetics (e.g. translocation through WGS), and conducted dimensionality reduction analysis/clustering to identify clusters of cells with similar multiomic profile.

- c. The focus of the investigation is a cluster of cells that is mainly composed of RMM cells (Figure 2c-e), which is proposed to correlate with: male, t(4;14), gain/amp1q21, del12p, mutations (NRAS, KRAS, DIS3, TP53, FGFR3, PRDM1 and NF1, Figure 2D).
- d. The authors propose that they have found a link between a 1q21 TF (PBX1) and PHF19, and that both are relevant to RRMM.

2. Questions to authors:

- a. For Figure 2A, what is the meaning of the violin plots? Are these expression or accessibility (promoter or around gene)? Please add labels/captions.
- b. For Figure 2B, it is very hard to see the different clusters, in C it is not possible to link disease state because it seems the cells of each state are spread everywhere. Perhaps a confusion matrix of clusters vs. samples with disease state? This could replace Figure 2E and clearly show how cluster 11 is mainly RRMM samples.
- a. What is the meaning of the numbers 23, 11, 20 and 22?
- b. Please clarify in text how Figure 2D was generated. More specifically, what is the meaning of the disks with different colors (red vs. blue). It appears to be log₂ fold-change but of what? For instance, what does it mean that cluster 11 has a -Log₁₀P of 5 and Log₂FC for SMM->RRMM?
- c. Similarly for Figure 2D, how is it possible to associate mutations with each cluster, given mutations are bulk and clusters are single cell? Same question for Figure 2G/2H. Are the authors using the same method as for Figure 2E? While all cells from each sample can be classified as the same disease state as the sample, the same is not true for mutations, cytogenetics, etc., as demonstrated by Figure 6.
- d. Please clarify the text from lines 161-163 in section "Identification of high-risk myeloma cells from scRNA-seq".
- e. Please clarify what test is described in line 238. Was it Fisher's exact test? Hypergeometric test? Why no multi-test correction?
- f. In lines 253-254, please confirm whether del17p was considered, given the two events often co-occur in MM.
- g. In lines 254-257 please correct/clarify the use of term synergy, which would mean higher effect than the sum of two events together. No data to support this claim is presented. Also, Figure 2HG should have 4 columns (missing gain/Amp1q21) or at least two different symbols in middle violin.
- h. Please clarify why multi-test correction was not used in line 269.
- i. Figure 3A's caption mentions 90 up- and 90-down regulated differentially expressed genes common between clusters 11, 20 and 22, but 3A appears to show fewer, perhaps only the up-regulated?
- j. It is stated that MKI67 and PHF19 co-express, but no correlation score is provided. Perhaps replace 3D by a regression line? Is this true in bulk RNA-seq? Is this true in all cells or just the ones in cluster 11?
- k. Please elaborate on the choice of pathways (GO vs. Hallmarks or KEGG) for Figure 3e-g. Please clarify the meaning of the color scale. It seems it is arbitrary for each of the 3 plots. Also, please confirm whether multi-test correction was performed and what cut-offs were used for these pathways. What is the meaning of x-axis (GeneRatio)? How is it possible to determine if these pathways are over- or under-expressed in these clusters? What enrichment algorithm was used (hypergeometric, FISHER's exact test)?
- l. Line 304, please explain test conducted (hypergeometric) and metrics (e.g. OR).
- m. For line 307, how about overlap between down-regulated by PHF19 KO and genes both down-regulated in ATAC/RNA-seq, as a negative control?
- n. Please label Figure 4A-B as accessibility/ATAC and gene expression/RNA-seq, respectively. Please conduct multi-test correction and use -Log₁₀q value instead.
- o. Please label other genes associated with proliferation in 4A-B, including when they are not statistically significant (e.g. in 4A there seem to be only 3, in 4B there are many).
- p. Lines 317/324:
- a. If the method used to associate gain/amp1q21 with clusters was increased proportion of cells from samples with gain/amp1q21 (please confirm by addressing prior questions) then, given the increased prevalence of gain/amp1q21 in RRMM patients, it is given that the cluster with most RRMM cells would also be associated with gain/amp1q21.

- b. It is reported that a list of 35+57 TFs were identified to interact with PHF19, but according to datasets such as Genecards, there are many more. While these are based on putative binding sites, the list may be larger than proposed by authors).
- c. Please conduct multi-test correction for lines 322-323.
- q. Figure 5A and B: It is confusing how the view of chromosome 1 is large, then below in very small case, chromosome 9, which is the most important for B. PHF19 gene and promoter are hidden below the U266 chip-seq peak, please move up close to chromosome distances in B.
- r. Not clear if 5C-D are necessary. Multi-test correction required for 5G.
- s. Lines 329-333 appear to be a self-fulfilling prophecy, given that they are a direct result from PHF19 being chosen because it was differentially expressed in RRPC11, and PBX1 because it was part of amplification enriched in this cluster, so it is not an independent biological link.
- t. While the anecdotal observations from section "Analysis of patient subclones[...]" are interesting, they are not validation of the hypotheses proposed by authors. These could be confirmed in bulk RNA-seq cohorts, such as Compass.
- u. Please clarify the putative link between PHF-19 and PBX1. Please see attached example from a cohort of MM patients in different disease states with bulk RNA-seq data. While there is a marked increase in expression in PHF19 (as previously proposed in DREAM challenge), there is no correlation with PBX1. The authors should be able to confirm these data with Compass dataset or bulk RNA-seq data from their own cohort.

3. Recommendation from reviewer:

- a. The following issues negatively affect the impact of this study:
 - i. No validation of findings in existing bulk RNA-seq cohorts (e.g. MMRF's Compass);
 - ii. No FISH data appears to have been used to validate any of the inferred CNV/translocations, why?
 - iii. Lax statistical rigor; for example, no multi-test corrections performed, loose use of terms such as synergy (line 393) colloquially.
 - iv. There does not seem to exist a new finding in the article, and instead the focus is on well-known genetic/cytogenetic markers (amp1q21/mut TP53), and gene PHF19, but no proposal of new mechanism;
- a. The cluster identified by this study is only as large as 15% of one RRMM, and as low as 5%, suggesting it may not be as important as the authors propose, in terms of refractory disease (Lines 309-311).
- a. Due to lack of correlation/validation with existing cohorts, it is not clear the relevance of the cluster of cells chosen by the authors to focus in this work.
- b. Unless it is possible to confirm genomic events in each individual cluster, the statistical test that this reviewer believes the authors are conducting to associate clusters with genomic/cytogenetic events, is incorrect. As demonstrated by multiple studies, including Figure 6, there is intra-sample heterogeneity, and the same sample may contain cells with and without mutations, in different clusters.
- b. In summary, while the authors propose a first of a kind dataset of scMultiomics data, the methods used to analyze these data appear to be incorrect/flawed or confused and unclear at times, and the lack of any validation against publicly existing data is worrisome. This reviewer suggests a major review of the article, preferentially adopting an agnostic approach, with more strict statistical plan.

GEP

Reviewer #3:

Remarks to the Author:

This study by Johnson et al. represents a comprehensive analysis of myeloma progression at the single-cell level, shedding light on the role of genetic and epigenetic factors in disease progression. The authors generated a large multi-omic single cell dataset consisting of a total of 325,025 cells from 49 patients, including smoldering multiple myeloma (SMM), newly diagnosed multiple myeloma (NDMM), and relapsed or refractory multiple myeloma (RRMM) patients, with matched genomic profiles. They investigated how cytogenetic changes, particularly Gain/Amp1q, correlate with molecular subtypes and clinical outcomes in myeloma. The authors identify distinct clusters of myeloma cells called relapsed/refractory plasma cells (RRPCs) associated with disease progression, which exhibit a higher proportion of Gain/Amp1q events, TP53 mutations, increased expression of the epigenetic regulator PHF19 and a high proliferative index.

PHF19 negatively regulates cell cycle inhibitor genes including CDKN1C, therefore promoting proliferation. This suggests a potential mechanism for how PHF19 contributes to disease progression. The study identifies candidate transcription factors, including PBX1, RFX5, and RBBP5, located on chromosome 1q, which may regulate PHF19 expression. Analysis of genetic subclones in patients further reveals that subpopulations with Gain/Amp1q exhibit increased PHF19 expression, therefore strengthening the link between Gain/Amp1q events and PHF19 expression. This also highlights the potential impact of subclonal differences in genetic events on disease progression.

The study provides valuable insights into the molecular mechanisms driving myeloma progression, especially in the context of Gain/Amp1q and PHF19 and helps explain why Gain/Amp1q is a high-risk cytogenetic event in myeloma. It addresses an important knowledge gap in understanding the interplay between genetic and epigenetic factors at the single-cell level.

The authors generated one of the largest single cell multi-omic datasets for myeloma which will be valuable to the community. The dataset appears to be of good quality. However, there are some outstanding issues that should be addressed where additional information or analyses should be provided. In some places, the analyses seem preliminary, and the authors could make more use of the fact that they generated large multi-omic single cell datasets to explore the epigenetic regulation of PHF19.

Major comments:

1. The link between PBX1 and PHF19 is weak considering that this represents one of the main novel findings of this study. The authors argue that PBX1 binds the PHF19 promoter based on PBX1 Chip-Seq peaks. While there seems to be a promoter peak in the MM1.S cell line (III MM1.S), in U266 the indicated peak appears to be intronic. While this peak might represent an enhancer, no evidence is provided that it regulates PHF19. Furthermore, the region shown in Fig. 5B represents an area largely downstream of PHF19. There are several ways in which the authors could explore the epigenetic regulation of PHF19 in greater detail. Given that the authors generated a large multi-omic single cell dataset, they should make use of this by predicting transcription factor activity using tools, such as SCENIC, to explore which TFs show the greatest specificity to the RRPC clusters. It would be important to include tracks from the scATAC-Seq showing the PHF19 locus comparing the RRPC clusters to other clusters to explore its regulation. Using peak-gene-linkage, they could link possible enhancer peaks to PHF19. The authors could further use the scATAC-Seq data for motif analysis and to estimate TF activity of PBX1, RFX5 and RBBP5 in the RRPC clusters. These analyses would provide further mechanistic insight into how PHF19 is regulated by Gain/Amp1q.
2. The study is correlative and to establish a causal link, some functional validation should be performed. The authors include RNA-Seq data following PBX1 kd showing that PHF19 is downregulated. Does PBX1 ko result in a decrease in proliferation for example?
3. In the analysis of genetic heterogeneity and subclones, the authors could further calculate cell cycle scores for each subclone to strengthen the link between proliferation, Gain/Amp1q and PHF19.
4. The authors should include details on quality control filtering performed and basic QC measures on

the dataset, both for scRNA- and scATAC-sequencing. Marker genes/biological processes for cluster 22 seem to be enriched for ribosomal and mitochondrial RNAs, which can be sign of poor-quality cells. Therefore, some QC should be shown.

5. Further details should be provided on some of the analyses. How was the association of particular cytogenetic events with clusters (shown in Table 3) determined? How were significant associations between clusters and covariates shown in Fig. 2D determined? How was statistical analysis performed (e.g. in Fig 2E-H)?

6. The role of Tp53 in this remains unclear.

Minor comments:

1. Some of the figures are hard to read. In Fig. 2B, 2D, 3E-G the size of the legends should be increased.

2. The authors should make the size Venn diagram in Fig. 4C-E proportional to the number of genes.

3. The gene and promoter region in Fig. 5B should be labelled.

REVIEWER COMMENTS

We thank the reviewers for their insightful feedback and appreciate how thoroughly they reviewed our manuscript. We believe that we have made all of the appropriate changes needed and think that our manuscript is even stronger now thanks to these reviews. To simplify the review of our revised manuscript we have made all comments to the reviewers in blue and marked all changes in the manuscript in red.

Reviewer #1, expertise in MM genomics and molecular pathogenesis (Remarks to the Author):

Understanding the pathogenesis and biology behind 1q amplification and its impact on myeloma cells is one of the most important questions in MM biology. This manuscript provides an exceptionally rich resource with WGS, RNAseq and single cell studies (ATAC and scRNAseq) to provide a comprehensive genomic dataset. The single cells studies are particularly important because they are novel, but also because they allow the dissection of subclonal heterogeneity that has been obscured in prior studies. Needless to say, synthesizing all of this diverse data into a comprehensible analysis is a huge challenge, which the authors have met admirably.

Most scRNA in MM has been on unselected BM cells, this is the first to focus on the heterogeneity within the tumor cell compartment. Important the cluster 11 (largest one associated with progression) increased from 1%>3%>8% with SMM>MM>RRMM. This is reminiscence of the plasma cell labelling index which (using much older technology) identified a similar fraction of proliferative cells. The big advantage now is that the genomic information for these cells is also captured, and can be analyzed.

The focus on PHF19 is appropriate, given it's outsize impact on prognosis. The functional studies show clear PHF19-dependent cell proliferation. Importantly they show that PHF19 is a PBX1 target gene providing one plausible pathogenic scenario in which 1q amplification leads to increased pBX1 expression which drives PHF19 and FOXM1 expression.

These studies (and those in reference 11) show MM growth is dependent on PBX1 and PHF19, the next step for future studies is to show that over-expression of these factors can accelerate MM growth, the requirement to consider these driver genes.

1. Fig S3. Not clear what numbers in parentheses represent.

We thank the reviewer for catching this and have clarified these numbers in the figure legend for Fig S3.

The mean expression in CPM of the marker genes are included in parentheses for *CDKN2C* (B), *CKS1B* (D), *RB1* (F), and *TP53* (H).

2. Not clear how del1p, gain1q, del13, del17p defined. Was CN of *CDKN2C*, *CKS1B*, *RB1* and *TP53* used? If so, list in the figures the gene name of the CN plotted (as they do in other figures). Provide the criteria they used to define these CAN.

We agree with the reviewer and have added additional clarification on which genes were used to call these CNAs and more specifics about the criteria used to call CNAs (lines 133-135).

Copy number variations were analyzed using ASCAT-NGS²⁵ and were defined as follows: 0 copies (deep deletion), 1 copy (deletion), 2 (normal), 3 (gain), and 4+ (amplification).

patient. **B)** Final clusters determined from the dataset integration of 49 samples. **C)** Diagnosis of the patient from which each cell was derived overlaid on to the integrated clusters. **D)** Significant associations between clusters and clinical/genomic covariates. See **Supplementary Materials** for additional information on how this figure panel was generated. **E)** Percentage of each cluster from SMM, NDMM, and RRMM patients. RRPC clusters are boxed in red (**D-E**). **F)** Proportion of RRPCs in patients stratified by diagnosis. **G)** Proportion of RRPC₁₁ in patients stratified by 1q copy number. **H)** Proportion of RRPC₁₁ in patients stratified by *TP53* mutation status. **I)** Proportion of RRPC₁₁ in patients with gain/*Amp1q*, *TP53* mutations, or both. Significance levels are: P<0.1: (·), P<0.05: *, P<0.01 (**), P<0.001 (***)

3. Fig 3A is hard to interpret. Presumably 1-25 refer to the clusters, and each vertical line is an individual patient. It is not clear if what is plotted is the number of cells which express the given marker, or the average expression of the marker among the patients cells in the cluster. It would be helpful to order the genes based on upregulation and downregulation in the high risk clusters. It is not clear to me that any genes are downregulated. Perhaps a different normalization would help with this.

We appreciate the reviewer noticing this and we did have an error in the figure legend. Downregulated genes are not included in the heatmap. We have also further explained the layout of the figure in the figure legend. See reviewer 1 comment 4 for complete figure and figure legend.

4. Fig 4D. The co-expression of PHF19 and MKI67 is not convincing. It looks like some cells are predominantly red, some predominantly green, and only 1 yellow (expressing both). I think an X-Y plot would be more convincing. I would expect that some patients may have biallelic deletions of *CDKN2C*, or *RB1* with high MKI67 without elevated PHF19, which might become more evident.

This is a good point and we agree that these panels were difficult to interpret. For these reasons, we smoothed the expression of PHF19 and MKI67, replotted the data, and included an x-y plot to show the correlative nature of these genes (see below).

Fig. 3 Differential expression for the top intersecting RRPC₁₁, RRPC₂₀, and RRPC₂₂ markers. **A)** Heatmap of the top 90 up-regulated DEGs common between RRPC₁₁, RRPC₂₀, and RRPC₂₂. The DEGs were sorted by their rank sum across RRPC₁₁, RRPC₂₀, and RRPC₂₂. **B)** RRPC clusters coexpress *PHF19* and *MKI67*. **C)** *PHF19* and *MKI67* expression are correlated in their expression. Note that expression values were smoothed using 20 nearest neighbors for visualization purposes in **B-C**. **D-F)** Top 15 significant GO Biological Process for RRPC₁₁ DEGs (**D**), RRPC₂₀ DEGs (**E**), and RRPC₂₂ DEGs (**F**).

Reviewer #2, expertise in MM omics and systems biology (Remarks to the Author):

1. Summary from reviewer:

- a. This article investigates data from of 49 scMultiomics (10 SMOL, 22 NDMM and 17 RMM), as well as 44 WGS samples.
- b. Initially, authors characterized each sample in terms of mutations, copy number, cytogenetics (e.g. translocation through WGS), and conducted dimensionality reduction analysis/clustering to identify clusters of cells with similar multiomic profile.
- c. The focus of the investigation is a cluster of cells that is mainly composed of RMM cells (Figure 2c-e), which is proposed to correlate with: male, t(4;14), gain/amp1q21, del12p, mutations (NRAS, KRAS, DIS3, TP53, FGFR3, PRDM1 and NF1, Figure 2D).
- d. The authors propose that they have found a link between a 1q21 TF (PBX1) and PHF19, and that both are relevant to RRMM.

2. Questions to authors:

- a. For Figure 2A, what is the meaning of the violin plots? Are these expression or accessibility (promoter or around gene)? Please add labels/captions.

We thank the reviewer for noticing this and we have added the additional required details to the figure legend.

Fig. 2 Characterization and integration of plasma cell clusters. A) Canonical translocations, mutations, and copy number alteration marker gene expression in each patient.

- b. For Figure 2B, it is very hard to see the different clusters, in C it is not possible to link disease state because it seems the cells of each state are spread everywhere. Perhaps a confusion matrix of clusters vs. samples with disease state? This could replace Figure 2E and clearly show how cluster 11 is mainly RRMM samples.

This is a good point and we agree that a heatmap would make it easier to see the high percentage of cluster 11 coming from RRMM patients. Due to this we have added an additional panel to Fig 2, panel E, so that the reader can easily see the high proportion of cluster 11 that is made up of mostly RRMM cells. We have also appropriately updated the figure legend. Please see reviewer 1 comment 2 for the updated Fig. 2 and figure legend.

- c. What is the meaning of the numbers 23, 11, 20 and 22?

We apologize for any confusion. The numbers indicate the clusters that make up the RRPCs or clusters that were significantly associated with RRMM per our significance testing in Fig 2D. In this manuscript we focus mainly on cluster 11 or (RRPC₁₁) because it is the largest of the RRPC subclusters (23, 11, 20, 22). See lines 241-250.

We determined that clusters 11, 20, 22, and 23 were associated with later stages of myeloma progression (**Fig. 2B-E**) such that all of these clusters had significantly greater proportions in RRMM compared to SMM (P<0.001 and FDR=0.003, P<0.001 and FDR=0.003, P=0.031 and FDR=0.24, and P=0.032 and FDR=0.25, respectively, **Table S4**). The largest of these clusters, cluster 11, increased in proportion from SMM to NDMM (1% to 3%, P=0.008 and FDR=0.13, **Fig. 2F, Table S4**), and NDMM to RRMM (3% to 8%, P<0.001 and FDR=0.03, **Fig. 2F, Table S4**). Due to their greater enrichment in RRMM patients in comparison to earlier stages of myeloma we denote clusters 11,

20, 22, and 23 as relapse/refractory plasma cells (RRPC) or RRPC₁₁, RRPC₂₀, RRPC₂₂, and RRPC₂₃ for individual RRPC clusters (**Table 3**).

d. Please clarify in text how Figure 2D was generated. More specifically, what is the meaning of the disks with different colors (red vs. blue). It appears to be log₂ fold-change but of what? For instance, what does it mean that cluster 11 has a -Log₁₀P of 5 and Log₂FC for SMM->RRMM?

We thank the reviewer for pointing out the missing methods for this figure panel and have included them in the supplementary materials and in the figure legend.

Figure Legend:

D) Significant associations between clusters and clinical/genomic covariates. See Supplementary Materials for additional information on how this figure panel was generated.

Supplementary Materials:

Calculating associations between clusters and covariates

We performed comprehensive hypothesis testing to evaluate whether clusters were enriched in specific genomic, cytogenetic, and clinical patient groups. Each pairwise comparison was evaluated for the proportion of clusters 1-25 and patient covariates: SMM vs RRMM, SMM vs NDMM, NDMM vs RRMM, age ≥ 65 vs age < 65, t(4;14) vs not t(4;14), t(11;14) vs not t(11;14), t(6;14) vs not t(6;14), t(14;16) vs not t(14;16), t(14;20) vs not t(14;20), HRD vs not HRD, t(MYC) vs not t(MYC), MYC abnormality vs no MYC abnormality, Del(CDKN2C) vs Norm(CDKN2C), Del(RPL5) vs Norm(RPL5), Del(TENT5C) vs Norm(TENT5C), Del(PRKN) vs Norm(PRKN), Del(DOCK5) vs Norm(DOCK5), Del(CDKN2A) vs Norm(CDKN2A), Del(BIRC3) vs Norm(BIRC3), Del(ATM) vs Norm(ATM), Del(CDKN1B) vs Norm(CDKN1B), Del(BRCA2) vs Norm(BRCA2), Del(RB1) vs Norm(RB1), Del(DIS3) vs Norm(DIS3), Del(ABCD4) vs Norm(ABCD4), Del(traf3) vs Norm(traf3), Del(CYLD) vs Norm(CYLD), Del(WWOX) vs Norm(WWOX), Del(MAF) vs Norm(MAF), Del(TP53) vs Norm(TP53), Gain/Amp(CKS1B) vs Norm(CKS1B), Amp(CKS1B) vs Norm(CKS1B), Gain/Amp(TNFAIP8) vs Norm(TNFAIP8), Amp(TNFAIP8) vs Norm(TNFAIP8), Gain/Amp(TNXB) vs Norm(TNXB), Amp(TNXB) vs Norm(TNXB), Gain/Amp(ADCY2) vs Norm(ADCY2), Amp(ADCY2) vs Norm(ADCY2), Gain/Amp(CKS1B) vs Norm(CKS1B), Amp(CKS1B) vs Norm(CKS1B), Gain/Amp(RAPGEF5) vs Norm(RAPGEF5), Amp(RAPGEF5) vs Norm(RAPGEF5), Gain/Amp(KLF14) vs Norm(KLF14), Amp(KLF14) vs Norm(KLF14), Gain/Amp(MYC) vs Norm(MYC), Amp(MYC) vs Norm(MYC), Gain/Amp(CDKN2A) vs Norm(CDKN2A), Amp(CDKN2A) vs Norm(CDKN2A), Gain/Amp(RNF20) vs Norm(RNF20), Amp(RNF20) vs Norm(RNF20), Gain/Amp(traf2) vs Norm(traf2), Amp(traf2) vs Norm(traf2), Gain/Amp(RRAS2) vs Norm(RRAS2), Amp(RRAS2) vs Norm(RRAS2), Gain/Amp(CCND1) vs Norm(CCND1), Amp(CCND1) vs Norm(CCND1), Gain/Amp(BIRC) vs Norm(BIRC), Amp(BIRC) vs Norm(BIRC), Gain/Amp(ATM) vs Norm(ATM), Amp(ATM) vs Norm(ATM), Gain/Amp(CDKN1B) vs Norm(CDKN1B), Amp(CDKN1B) vs Norm(CDKN1B), Gain/Amp(DNMT3A) vs Norm(DNMT3A), Amp(DNMT3A) vs Norm(DNMT3A), Gain/Amp(CRBN) vs Norm(CRBN), Amp(CRBN) vs Norm(CRBN), Gain/Amp(FGFR3) vs Norm(FGFR3), Amp(FGFR3) vs Norm(FGFR3), Gain/Amp(ABCD4) vs Norm(ABCD4), Amp(ABCD4) vs Norm(ABCD4), Gain/Amp(traf3) vs Norm(traf3), Amp(traf3) vs Norm(traf3), Gain/Amp(WDR72) vs Norm(WDR72), Amp(WDR72) vs Norm(WDR72), Gain/Amp(BLM) vs Norm(BLM), Amp(BLM) vs Norm(BLM), Gain/Amp(AKAP1) vs Norm(AKAP1), Amp(AKAP1) vs Norm(AKAP1), Gain/Amp(ZNF426) vs Norm(ZNF426), Amp(ZNF426) vs Norm(ZNF426), Gain/Amp(ZNF227) vs Norm(ZNF227), Amp(ZNF227) vs Norm(ZNF227), Gain/Amp(SON) vs Norm(SON), Amp(SON) vs Norm(SON), Mut(KRAS) vs not Mut(KRAS), Mut(NRAS) vs not Mut(NRAS), Mut(TENT5C) vs not Mut(TENT5C), Mut(DIS3) vs

not Mut(*DIS3*), Mut(*BRAF*) vs not Mut(*BRAF*), Mut(*TRAF3*) vs not Mut(*TRAF3*), Mut(*TP53*) vs not Mut(*TP53*), Mut(*CYLD*) vs not Mut(*CYLD*), Mut(*MAX*) vs not Mut(*MAX*), Mut(*UBR5*) vs not Mut(*UBR5*), Mut(*USP7*) vs not Mut(*USP7*), Mut(*IRF4*) vs not Mut(*IRF4*), Mut(*SP140*) vs not Mut(*SP140*), Mut(*PTPN11*) vs not Mut(*PTPN11*), Mut(*RB1*) vs not Mut(*RB1*), Mut(*NFKBIA*) vs not Mut(*NFKBIA*), Mut(*RASA2*) vs not Mut(*RASA2*), Mut(*TGDS*) vs not Mut(*TGDS*), Mut(*CDKN1B*) vs not Mut(*CDKN1B*), Mut(*DUSP2*) vs not Mut(*DUSP2*), Mut(*KLHL6*) vs not Mut(*KLHL6*), Mut(*EGR1*) vs not Mut(*EGR1*), Mut(*FGFR3*) vs not Mut(*FGFR3*), Mut(*ACTG1*) vs not Mut(*ACTG1*), Mut(*ZNF292*) vs not Mut(*ZNF292*), Mut(*HUWE1*) vs not Mut(*HUWE1*), Mut(*CCND1*) vs not Mut(*CCND1*), Mut(*SAMHD1*) vs not Mut(*SAMHD1*), Mut(*ABCF1*) vs not Mut(*ABCF1*), Mut(*CDKN2C*) vs not Mut(*CDKN2C*), Mut(*FUBP1*) vs not Mut(*FUBP1*), Mut(*PRDM1*) vs not Mut(*PRDM1*), Mut(*KMT2B*) vs not Mut(*KMT2B*), Mut(*ATM*) vs not Mut(*ATM*), Mut(*KMT2C*) vs not Mut(*KMT2C*), Mut(*CREBBP*) vs not Mut(*CREBBP*), Mut(*ARID1A*) vs not Mut(*ARID1A*), Mut(*ATRX*) vs not Mut(*ATRX*), Mut(*NF1*) vs not Mut(*NF1*), Mut(*TET2*) vs not Mut(*TET2*), Mut(*KDM5C*) vs not Mut(*KDM5C*), Mut(*ARID2*) vs not Mut(*ARID2*), Mut(*DNMT3A*) vs not Mut(*DNMT3A*), Mut(*KDM6A*) vs not Mut(*KDM6A*), and Mut(*MAFB*) vs not Mut(*MAFB*).

For each comparison (i) listed above, the mean proportion (Prop) of cluster j in across each patient was compared between the groups (Group) using a one sided t-test. For each combination of comparison (i) and cluster (j) a T, P, -Log₁₀P, and Log₂FC, were calculated based on pairwise t-tests.

$$T_{ij}, P_{ij} = t - test(Prop_j, Group_i)$$

To visualize these results the Log₂FC was used color the dots in our dot plot such that positive fold changes, i.e. there is a higher proportion in cluster j in the first listed group from comparison i. show a positive association between the covariate and cluster. The -Log₁₀P was used to scale the size of each dot so that more significant cluster proportion-covariate comparison had larger dots (**Fig. 2D**). Only covariates with at least one significant P were retained in the dot plot (**Fig. 2D**). The Benjamini-Hochberg false discovery rate (FDR) was also calculated for all of these comparisons. We considered a comparison significant if it's $P \leq 0.05$ and $FDR \leq 0.25$.

e. Similarly for Figure 2D, how is it possible to associate mutations with each cluster, given mutations are bulk and clusters are single cell? Same question for Figure 2G/2H. Are the authors using the same method as for Figure 2E? While all cells from each sample can be classified as the same disease state as the sample, the same is not true for mutations, cytogenetics, etc., as demonstrated by Figure 6.

This is a good point by the reviewer and as such we have added some more discussion about this in the Discussion section. From WGS, we can accurately calculate copy number, mutational, and translocation information at the patient level. We then can compare the proportion of a patient's sample in each cluster between the patient level WGS information. Please see the added discussion below (lines 407-412).

Besides the clear association of RRPCs with myeloma progression, an additive association with *TP53 mutations* and 1q CNVs was identified in these RRPCs. It is worth noting that we could not call mutations at the single cell level accurately using this type of single cell data. For these reasons we use the patient level mutational data from paired WGS to make inferences about the clusters. Once better methods become available we will call mutations at the single cell level using either RNA or ATAC reads.

f. Please clarify the text from lines 161-163 in section “Identification of high-risk myeloma cells from scRNA-seq”.

Thank you again for the feedback. We have added some additional clarification to this section (lines 163-169).

For each of the identified plasma cell clusters derived from the CD138+ samples, the proportion of that cluster was calculated for each patient such that each patient had a total proportion of 1.0 split between each individual cluster. For example, if a patient had 1000 cells where 500 cells were in cluster 1, 200 cells were in cluster 2, and 300 cells were in cluster 3, the cluster proportions for that patient would be 0.5 cluster 1, 0.2 cluster 2, and 0.3 cluster 3. This was repeated for each patient using the clusters from the integrated Seurat object.

g. Please clarify what test is described in line 238. Was it Fisher’s exact test? Hypergeometric test? Why no multi-test correction?

We appreciate the the reviewers insight on the statistical evaluation used in this study.

We have added a section in the supplementary materials entitled: “**Calculating associations between clusters and covariates**” where we describe in detail the statistics used for these tests. We have also included the Benjamini-Hochberg corrected values. The section mentioned here was also edited accordingly (lines 241-245).

We determined that clusters 11, 20, 22, and 23 were associated with later stages of myeloma progression (**Fig. 2B-E**) such that all of these clusters had significantly greater proportions in RRMM compared to SMM ($P < 0.001$ and $FDR = 0.003$, $P < 0.001$ and $FDR = 0.003$, $P = 0.031$ and $FDR = 0.24$, and $P = 0.032$ and $FDR = 0.25$, respectively, **Table S4**).

h. In lines 253-254, please confirm whether del17p was considered, given the two events often co-occur in MM.

We thank the reviewer for their comment. This del17p was considered and that information is included in the supplementary materials Table S4 (lines 260-262).

In addition, samples with a *TP53* mutation had a significantly greater proportion of RRPC₁₁ than samples without a *TP53* mutation (0.08 vs. 0.03, $P = 0.025$ and $FDR = 0.22$; **Fig. 2H, Table S4**).

i. In lines 254-257 please correct/clarify the use of term synergy, which would mean higher effect than the sum of two events together. No data to support this claim is presented. Also, Figure 2HG should have 4 columns (missing gain/Amp1q21) or at least two different symbols in middle violin.

This is a good suggestion and we have clarified this in the text and updated Fig 2. We have replaced the term “synergy” with “additive” because we aimed to show that the effect of *TP53* mutations and 1q cnvs were not entirely dependent on one another (lines 262-264).

There was also an additive effect between *TP53* mutation status and Gain/Amp1q such that the number of these alterations was correlated with RRPC₁₁ proportion ($PCC = 0.42$, $P = 0.004$, **Fig. 2I**).

j. Please clarify why multi-test correction was not used in line 269.

We are sorry for the confusion. We did apply Benjamini-Hochberg correction to the differential gene expression analysis and have updated the y axis label in Fig 4.

Fig. 4 Differential gene expression and chromatin accessibility in RRPC₁₁. **A)** Gene activity differential chromatin accessibility for RRPC₁₁ with all proliferation genes annotated. **B)** Differentially expressed genes for RRPC₁₁ with all proliferation genes annotated. **C)** Intersection of PHF19 repressed genes from Ren et al. versus RRPC₁₁ ATAC and RNA downregulated genes. **D)** Intersection of PHF19 up-regulated genes from Ren et al. versus RRPC₁₁ ATAC and RNA up-regulated genes. **E)** Intersection of **(D)** with proliferation signature genes.

k. Figure 3A's caption mentions 90 up- and 90-down regulated differentially expressed genes common between clusters 11, 20 ad 22, but 3A appears to show fewer, perhaps only the up-regulated?

We appreciate both reviewer 1 and reviewer 2 catching this mistake. These are the 90 upregulated genes. We have corrected this in the figure legend.

Figure legend:

A) Heatmap of the top 90 up-regulated DEGs common between RRPC₁₁, RRPC₂₀, and RRPC₂₂. The DEGs were sorted by their rank sum across RRPC₁₁, RRPC₂₀, and RRPC₂₂.

I. It is stated that MKI67 and PHF19 co-express, but no correlation score is provided. Perhaps replace 3D by a regression line? Is this true in bulk RNA-seq? Is this true in all cells or just the ones in cluster 11?

We appreciate both reviewer 1 and 2 catching this and we have improved our visualization of their coexpression. Specifically, we smoothed the expression using kNN smoothing, which is widely used in scRNA-seq analysis, then we replotted the scatterplot and added an x-y plot showing their correlation (see reviewer 1 comment 4 for the updated figure). The correlation that we reported is across all clusters. Furthermore, we have performed further testing in the NDMM patients from the MMRF CoMMpass study and found strong correlations between MKI67 and PHF19. Please see the below added text (lines 281-297).

Notably, RRPC_{11,20,22} had greater expression of *PHF19* than other clusters (RRPC₁₁: log₂FC=2.43, P<0.001 and FDR<0.001; RRPC₂₀: log₂FC =3.08, P<0.001 and FDR<0.001; RRPC₂₂: log₂FC=2.77, P<0.001 and FDR<0.001, **Fig. 3A+B**), an epigenetic modifier whose expression is highly predictive of poor prognosis in myeloma patients.¹³ One of the main markers for proliferation, *MKI67* is also highly expressed in RRPC_{11,20,22} (RRPC₁₁: log₂FC=3.78, P<0.001 and FDR<0.001; RRPC₂₀: log₂FC=4.37, P<0.001 and FDR<0.001; RRPC₂₂: log₂FC=3.80, P<0.001 and FDR<0.001, **Fig. 3A,C**) and co-expressed with *PHF19* (PCC=0.37, P<0.001, **Fig. 3D**). This correlation between *MKI67* and *PHF19* expression can also be identified in NDMM patients from the MMRF CoMMpass study (**Fig. S6, Table S9+S10**). In the CoMMpass NDMM cohort, *MKI67* and *PHF19* expression (Log₂TPM) were highly correlated (PCC=0.70, P<0.001 and FDR<0.001, **Table S9**). In the CoMMpass NDMM cohort Amp1q, Gain1q, and Normal 1q subpopulation, their PCC values were also high (PCC=0.83, PCC=0.65, PCC=0.70, P<0.001 and FDR<0.001, P<0.001 and FDR<0.001, P<0.001 and FDR<0.001, respectively, **Table S9**). Both *MKI67* and *PHF19* showed significantly over-expression in Amp1q vs. normal 1q (Log₂FC=0.49, Log₂FC=0.52, P<0.001 and FDR<0.001, P=0.002 and FDR=0.003, respectively, **Table S10**).

Fig. S6 Association between MKI67 and PHF19 expression in the MMRF CoMMpass study cohort of patients. The points in the plot are labeled by their CNV status for 1q (*CKS1B*). The trend lines are fit for each individually.

Table S7 Association between *PHF19* and *MKI67* expression (Log_2TPM) in the MMRF CoMMpass cohort of NDMM patients.

Cohort	PCC	PCC P	FDR	SCC	SCC P	FDR
NDMM Amp1q	0.83	9.36E-11	9.36E-11	0.77	9.02E-09	9.02E-09
NDMM Gain1q	0.65	1.67E-23	2.23E-23	0.66	6.63E-24	8.84E-24
NDMM Norm 1q	0.70	9.11E-60	1.82E-59	0.65	9.22E-49	1.84E-48
NDMM (All patients)	0.70	3.99E-92	1.60E-91	0.67	2.02E-81	8.08E-81

Table S8 Differential expression of MKI67 and PHF19 expression (Log₂TPM) in the MMRF CoMMpass cohort of NDMM patients.

Gene	Group 1	Group 2	log2FC (Group1/Group2)	P	FDR
MKI67	Amp1q	Gain1q	0.27	2.78E-02	3.33E-02
MKI67	Amp1q	Norm1q	0.49	2.03E-04	6.08E-04
MKI67	Gain1q	Norm1q	0.21	9.76E-04	1.95E-03
PHF19	Amp1q	Gain1q	0.16	2.80E-01	2.80E-01
PHF19	Amp1q	Norm1q	0.52	2.42E-03	3.63E-03
PHF19	Gain1q	Norm1q	0.36	1.81E-04	6.08E-04

m. Please elaborate on the choice of pathways (GO vs. Hallmarks or KEGG) for Figure 3e-g. Please clarify the meaning of the color scale. It seems it is arbitrary for each of the 3 plots. Also, please confirm whether multi-test correction was performed and what cut-offs were used for these pathways. What is the meaning of x-axis (GeneRatio)? How is it possible to determine if these pathways are over- or under-expressed in these clusters? What enrichment algorithm was used (hypergeometric, FISHER's exact test)? We thank the reviewer for pointing out that there are also other ontologies available for functional enrichment analysis. To keep our analysis consistent, we only included GO Biological Processes in our analysis because it is a very commonly used ontology and because it contains terms related to processes like proliferation. We have added additional explanation of the figure in the supplementary materials (see below). We also updated the figure such that the $-\text{Log}_{10}(\text{P})$ are now $-\text{Log}_{10}(\text{FDR})$ reflecting that the values in the plots are already BH corrected. Please see Reviewer 1 comment 4 for the updated Fig. 3 and figure legend.

SUPPLEMENTARY METHODS:

Enrichment Analysis

To better understand the biology underlying different subsets of cells from the clustering results, we performed gene set enrichment analysis for the RRPC clusters. The DEG markers for each of the RRPC clusters were used as input to the clusterProfiler program to identify enriched Gene Ontology (GO) terms. The enrichGO function uses a hypergeometric test to identify GO gene sets that overlap significantly with a set of DEGs. These terms were then displayed as a dot plot to show the number of genes and the significance level of each term. The GeneRatio denotes the percentage of the DEGs that are also contained in a specific GO term. The Count variable denotes the number of genes in the numerator of the GeneRatio, i.e. the number of DEGs also contained in the GO term.

Aside from studying the functional enrichment of our DEGs with GO terms, it was also important to study the overlap of DEGs with other gene sets from previous studies. For these purposes, we utilized Fisher's exact test to calculate the P and also we used the Odds Ratio (OR) to study the effect size.

n. Line 304, please explain test conducted (hypergeometric) and metrics (e.g. OR). We appreciate the feedback and have clarified which test was used in the manuscript. Please see reviewer 2 comment m for the text that was added.

o. For line 307, how about overlap between down-regulated by PHF19 KO and genes both down-regulated in ATAC/RNA-seq, as a negative control?

This is a great point and we performed the suggested analysis as a negative control. Of the PHF19 KO down-regulated and Rescue up-regulated genes from Ren et al. there was only one gene overlap with the RNA and ATAC down-regulated from our single cell study. This is in contrast to the significant overlap of 13 genes for the RNA and ATAC up-regulated genes. Of the PHF19 KO up-regulated and Rescue down-regulated genes from Ren et al. there were only 18 genes overlapping with the RNA and ATAC up-regulated from our single cell study. This is in contrast to the significant overlap of 55 genes for the RNA and ATAC down-regulated genes. Neither of these comparisons were significant at $\alpha=0.05$ (lines 327-332).

In contrast neither of the opposite comparisons were significant. Genes that were up-regulated from the *PHF19* knockdown experiments and genes upregulated in both the ATAC-seq and RNA-seq from RRPC₁₁ had little overlap (OR=0.76, P=0.33). Genes that were down-regulated by *PHF19* knockdown had insignificant overlap with genes found to be downregulated in both ATAC-seq and RNA-seq from RRPC₁₁ (OR=0.18, P=0.067).

p. Please label Figure 4A-B as accessibility/ATAC and gene expression/RNA-seq, respectively. Please conduct multi-test correction and use -Log₁₀q value instead. We appreciate the feedback and have added this suggestion into our figure. See reviewer 2 comment j for the updated figure.

q. Please label other genes associated with proliferation in 4A-B, including when they are not statistically significant (e.g. in 4A there seem to be only 3, in 4B there are many). We have included all proliferation genes. See reviewer 2 comment j for the updated figure.

r. Lines 317/324:

i) If the method used to associate gain/amp1q21 with clusters was increased proportion of cells from samples with gain/amp1q21 (please confirm by addressing prior questions) then, given the increased prevalence of gain/amp1q21 in RRMM patients, it is given that the cluster with most RRMM cells would also be associated with gain/amp1q21.

We thank the reviewer for pointing this out. GainAmp1q21 will be generally correlated with RRMM. However, we are studying the mechanisms underlying the RRMM cells. For these reasons, it is still useful to identify which cytogenetic events are correlated with our cell clusters beyond their stage of disease. For these reasons, we not only identify clusters that are associated with disease stage but also with the underlying genetic events that are driving the myeloma cells into a progressive state.

ii) It is reported that a list of 35+57 TFs were identified to interact with PHF19, but according to datasets such as Genecards, there are many more. While these are based on putative binding sites, the list may be larger than proposed by authors).

We thank the reviewer for pointing this out and appreciate that there are more resources available to identify other TFs. Aggregating all possible transcription factors from all possible databases is a project by itself. For these reasons, we selected two highly cited TF databases for use in this analysis. For future work, it would be useful to aggregated all TF databases into a single source that could be used by the community.

iii) Please conduct multi-test correction for lines 322-323.

We thank the reviewer for their feedback and have added the multiple testing correction (lines 345-349).

Of these TFs, seven (*ATF3*, *KDM5B*, *PBX1*, *RBBP5*, *RFX5*, *USF1*, *ZNF648*) were located on 1q, of which *PBX1* ($\log_2FC=0.78$, $P < 0.001$ and $FDR < 0.001$), *RFX5* ($\log_2FC=0.81$, $P < 0.001$ and $FDR < 0.001$), and *RBBP5* ($\log_2FC=0.93$, $P < 0.001$ and $FDR < 0.001$) were significantly up-regulated in RRPC₁₁ compared to all other clusters (**Fig. 5A**).

s. Figure 5A and B: It is confusing how the view of chromosome 1 is large, then below in very small case, chromosome 9, which is the most important for B. PHF19 gene and promoter are hidden below the U266 chip-seq peak, please move up close to chromosome distances in B.

We are sorry for any confusion. Fig. 5A is meant to show the locations of the prospective TFs on chromosome 1q and Fig 5B is meant to show the PBX1 ChIP-seq peaks upstream of PHF19.

t. Not clear if 5C-D are necessary. Multi-test correction required for 5G.

We appreciate you noticing this mistake in the figure. We had already performed BH correction on these p-values and have updated the figure with the FDR labelled in Fig. 5G. We still think that it is relevant to include Fig. 5C-D because they show visually the relative rank of the peak strength.

u. Lines 329-333 appear to be a self-fulfilling prophecy, given that they are a direct result from PHF19 being chosen because it was differentially expressed in RRPC₁₁, and PBX1 because it was part of amplification enriched in this cluster, so it is not an independent biological link.

We are sorry for any confusion and have edited these sections to clarify these results more. From the DREAM challenge and our previous work, we know that PHF19 is a high-risk marker for myeloma. We also know that Gain/Amp1q is a high-risk marker of myeloma and correlated with PHF19 expressing cells from our single cell analysis. We wanted to see if there were TFs located on 1q that could be regulating PHF19 and as a result produce more proliferative cell states. We have added the text below to describe these points (lines 361-370).

In our scRNA-seq, both *PHF19* ($\log_2FC=2.43$ $P < 0.001$ and $FDR < 0.001$) and *PBX1* ($\log_2FC=0.78$, $P < 0.001$ and $FDR < 0.001$) are upregulated in RRPC₁₁ (**Fig. 5G**). At the individual cell level, cells with Gain/Amp1q(*PBX1*) and Amp1q(*PBX1*) had increased expression of *PHF19* ($\text{Log}_2FC(\text{CPM})=0.94$, $P < 0.001$ and $\text{Log}_2FC(\text{CPM})=1.51$, $P < 0.001$, respectively). *PHF19* expression is detectable in 47% of RRPC₁₁ cells compared to only 9% of cells in other clusters and *PBX1* expression is detectable in 22% of RRPC₁₁ compared to only 10% of cells in other clusters, further strengthening their biological link. When we compared the correlation between PBX1 and PHF19 in the cohort of NDMM patients from the MMRF CoMMpass study we again identified a significant correlation ($PCC=0.21$, $P < 0.001$).

v. While the anecdotal observations from section “Analysis of patient subclones[...]” are interesting, they are not validation of the hypotheses proposed by authors. These could be confirmed in bulk RNA-seq cohorts, such as Compass.

We appreciate the comment and have conducted additional experiments using our scRNA-seq data and in CoMMpass to study PHF19, MKI67, Gain/Amp1q, and PBX1. Please see reviewer 2 comment l, reviewer 2 comment u, and below (lines 368-370).

When we compared the correlation between PBX1 and PHF19 in the cohort of NDMM patients from the MMRF CoMMpass study we again identified a significant correlation (PCC=0.21, P<0.001).

w. Please clarify the putative link between PHF19 and PBX1. Please see attached example from a cohort of MM patients in different disease states with bulk RNA-seq data. While there is a marked increase in expression in PHF19 (as previously proposed in DREAM challenge), there is no correlation with PBX1. The authors should be able to confirm these data with Compass dataset or bulk RNA-seq data from their own cohort.

We appreciate these suggestions to use outside datasets to validate the correlations that we have found in the single cell data. Using the CoMMpass dataset we again identified a correlation between PBX1 and PHF19 (see reviewer 2 comment v).

3. Recommendation from reviewer:

a. The following issues negatively affect the impact of this study:

i. No validation of findings in existing bulk RNA-seq cohorts (e.g. MMRF’s Compass);

This was a great point and as such we have provided more validation of our findings in the CoMMpass study. Please see comments above.

ii. No FISH data appears to have been used to validate any of the inferred CNV/translocations, why?

We appreciate the reviewer’s concerns. We included extensive WGS analysis in this study to get higher resolution CNV and translocation information. Due to the inclusion of the WGS data, we did not consider FISH in the analysis.

iii. Lax statistical rigor; for example, no multi-test corrections performed, loose use of terms such as synergy (line 393) colloquially.

We appreciate the reviewer pointing this out. We have included additional details of our statistical analysis, multiple testing correction, and additional statistical analyses.

iv. There does not seem to exist a new finding in the article, and instead the focus is on well-known genetic/cytogenetic markers (amp1q21/mut TP53), and gene PHF19, but no proposal of new mechanism;

We have tried to improve the writing and the discussion of this manuscript to make our point more apparent. The primary finding in this study is that PHF19 may be regulated by TFs on 1q leading to a more proliferative myeloma cell state. In addition, we have identified molecularly distinct clusters of myeloma cells and associated them with high-risk genomic events and chromatin states. We have found that only subsets of myeloma cells express high levels of PHF19 and reduce chromatin accessibility CDKN1C. This could especially useful since this subset of cells could eventually be targeted.

b. The cluster identified by this study is only as large as 15% of one RRMM, and as low as 5%, suggesting it may not be as important as the authors propose, in terms of refractory disease (Lines 309-311).

This is a great point. We are continuing to study this subset of cells considering they are enriched for high-risk markers and have a more proliferative state. It is worth noting that small subsets of cells can have an outsized impact on disease. Rare cell types may have a function in cellular communication, immune evasion, or act as a reservoir for disease during therapy leading to relapse. We already have additional experiments planned to phenotype these cells in more detail which we will include in future studies.

c. Due to lack of correlation/validation with existing cohorts, it is not clear the relevance of the cluster of cells chosen by the authors to focus in this work.

We thank the reviewer noticing this and have added more validation experiments using the MMRF CoMMpass dataset.

d. Unless it is possible to confirm genomic events in each individual cluster, the statistical test that this reviewer believes the authors are conducting to associate clusters with genomic/cytogenetic events, is incorrect. As demonstrated by multiple studies, including Figure 6, there is intra-sample heterogeneity, and the same sample may contain cells with and without mutations, in different clusters.

This is a good point and we are currently working on these analyses. It is not very straightforward to call mutations from single cell data because most modern scRNA-seq use RNA tagging techniques (10x Genomics). This means that SNVs can only be called from limited regions of genes near where they are tagged. Primary translocations proceed other genomic changes so there will likely not be major subclonal differences in translocations. For these reasons we have focused in this study on the CNVs and have focused at the patient level (WGS) for mutation information and translocations.

e. In summary, while the authors propose a first of a kind dataset of scMultiomics data, the methods used to analyze these data appear to be incorrect/flawed or confused and unclear at times, and the lack of any validation against publicly existing data is worrisome. This reviewer suggests a major review of the article, preferentially adopting an agnostic approach, with more strict statistical plan.

We appreciate all of the reviewers feedback and guidance and think that after addressing the many suggestions by the reviewer, we have made the manuscript more clear and the analysis plan more rigorous. We thank the reviewer for spending the time to provide many excellent suggestions for our manuscript.

Reviewer #3, expertise in scRNA-seq/scATACseq and MM (Remarks to the Author):

This study by Johnson et al. represents a comprehensive analysis of myeloma progression at the single-cell level, shedding light on the role of genetic and epigenetic factors in disease progression. The authors generated a large multi-omic single cell dataset consisting of a total of 325,025 cells from 49 patients, including smoldering multiple myeloma (SMM), newly diagnosed multiple myeloma (NDMM), and relapsed or

refractory multiple myeloma (RRMM) patients, with matched genomic profiles. They investigated how cytogenetic changes, particularly Gain/Amp1q, correlate with molecular subtypes and clinical outcomes in myeloma. The authors identify distinct clusters of myeloma cells called relapsed/refractory plasma cells (RRPCs) associated with disease progression, which exhibit a higher proportion of Gain/Amp1q events, TP53 mutations, increased expression of the epigenetic regulator PHF19 and a high proliferative index.

PHF19 negatively regulates cell cycle inhibitor genes including CDKN1C, therefore promoting proliferation. This suggests a potential mechanism for how PHF19 contributes to disease progression. The study identifies candidate transcription factors, including PBX1, RFX5, and RBBP5, located on chromosome 1q, which may regulate PHF19 expression. Analysis of genetic subclones in patients further reveals that subpopulations with Gain/Amp1q exhibit increased PHF19 expression, therefore strengthening the link between Gain/Amp1q events and PHF19 expression. This also highlights the potential impact of subclonal differences in genetic events on disease progression.

The study provides valuable insights into the molecular mechanisms driving myeloma progression, especially in the context of Gain/Amp1q and PHF19 and helps explain why Gain/Amp1q is a high-risk cytogenetic event in myeloma. It addresses an important knowledge gap in understanding the interplay between genetic and epigenetic factors at the single-cell level.

The authors generated one of the largest single cell multi-omic datasets for myeloma which will be valuable to the community. The dataset appears to be of good quality. However, there are some outstanding issues that should be addressed where additional information or analyses should be provided. In some places, the analyses seem preliminary, and the authors could make more use of the fact that they generated large multi-omic single cell datasets to explore the epigenetic regulation of PHF19.

Major comments:

1. The link between PBX1 and PHF19 is weak considering that this represents one of the main novel findings of this study. The authors argue that PBX1 binds the PHF19 promoter based on PBX1 Chip-Seq peaks. While there seems to be a promoter peak in the MM1.S cell line (III MM1.S), in U266 the indicated peak appears to be intronic. While this peak might represent an enhancer, no evidence is provided that it regulates PHF19. Furthermore, the region shown in Fig. 5B represents an area largely downstream of PHF19. There are several ways in which the authors could explore the epigenetic regulation of PHF19 in greater detail. Given that the authors generated a large multi-omic single cell dataset, they should make use of this by predicting transcription factor activity using tools, such as SCENIC, to explore which TFs show the greatest specificity to the RRPC clusters. It would be important to include tracks from the scATAC-Seq showing the PHF19 locus comparing the RRPC clusters to other clusters to explore its regulation. Using peak-gene-linkage, they could link possible enhancer peaks to PHF19. The authors could further use the scATAC-Seq data for motif

analysis and to estimate TF activity of PBX1, RFX5 and RBBP5 in the RRPC clusters. These analyses would provide further mechanistic insight into how PHF19 is regulated by Gain/Amp1q.

We thank the reviewer for their comment and have included more data to support the regulatory and epigenetic links between PBX1 and PHF19. From these same data we found more supporting evidence for PRC2 repression of CDKN1C too (lines 314-319, 355-356, 357-361).

RRPCs have reduced expression of cell-cycle inhibitors associated with loss of chromatin accessibility:

Since, previous studies have already established epigenetic regulatory mechanisms of *CDKN1C* by *PHF19*¹⁴ and *CDKN1C* is contained within a repressed polycomb regulatory region (Fig. S8A)³³, these results demonstrate that a subset of proliferative cells with high *PHF19* expression, found primarily in relapsed or refractory patients, also likely epigenetically downregulate *CDKN1C* (Fig. 4A+B).

PBX1 on 1q regulates expression of *PHF19* in RRPCs:

These peaks were also located in super-enhancer region within the *PHF19* genomic loci (Fig. 5B, Fig. S8B)³³.

Fig. S8 Regulatory regions determined from BLUEPRINT epigenomic data (GSE151556). **A**) CDKN1C has repressive regions associated with polycomb repressive complex 2 (H3K27me3). **B**) PHF19 has active regions representing a super-enhancer.

SUPPLEMENTARY METHODS:

Analysis of pre-B cell ALL data

Preprocessed ChIP-seq and RNA-seq data was downloaded from GSE138031. Specifically, MACS peak call BED and visualized peaks TDF files were downloaded for both PBX1 ChIP-seq and E2A/PBX1 ChIP-seq. The RNA-seq RPKM values were also downloaded for Scramble, E2A, and PBX1 shRNA (N=2 per group). Based on these data, the relative rank of peaks upstream of PHF19 were evaluated and the expression of PHF19 was compared between shRNA treated groups using a one-sided t-tests.

PBX1 on 1q regulates expression of PHF19 in RRPCs:

Similar patterns were also found in pre-B cell acute lymphoblastic leukemia (ALL) cell line (697)³⁴. In the pre-B ALL 697 cells, PBX1 ChIP-seq peaks existed upstream of PHF19 (**Fig. S9A-C**) and cells treated with shRNA silencing PBX1 had significantly decreased PHF19 expression ($P = 0.043$, **Fig. S9D**).

Fig. S9 Association of the chromosome 1q TF, PBX1, with *PHF19* expression in pre-B cell ALL cell line 697 (GSE138031). **A)** PBX1 ChIP-seq peaks upstream of *PHF19* on chromosome 9. **B-C)** PBX1 (**B**) and E2A/PBX1 (**C**) ChIP-seq peak scores in pre-B ALL 697 cells with the percentile of each ChIP-seq peak marked. **D)** Differences in *PHF19* expression in pre-B ALL 697 cells treated with PBX1 or E2A shRNAs. Significance levels are: P<0.1: (·), P<0.05: *, P<0.01 (**), P<0.001 (***).

2. The study is correlative and to establish a causal link, some functional validation should be performed. The authors include RNA-Seq data following PBX1 kd showing that PHF19 is downregulated. Does PBX1 ko result in a decrease in proliferation for example?

We thank the reviewer for their insight and yes PBX1 KO does result in a decrease in proliferation per the PBX1 study that we utilized in in Fig. 5 of our study. We have added additional discussion of this in the manuscript (lines 349-352).

PBX1 silencing experiments that resulted in reduced live cell percentage and lower tumor weight also had ChIP-seq and RNA-seq data available¹¹. This dataset was used to investigate PBX1 binding in myeloma cell lines.

3. In the analysis of genetic heterogeneity and subclones, the authors could further calculate cell cycle scores for each subclone to strengthen the link between proliferation, Gain/Amp1q and PHF19.

This is a great suggestion and we have included the cell cycle analysis results for clusters and subclones. See below (lines 298-303, 380-393).

RRPCs have increased expression of the epigenetic modifier *PHF19* and proliferative genes:

Furthermore, we see enrichment for cell-cycle related processes in RRPC₁₁ (**Fig. 3E, Fig. S7A**), RRPC₂₀ (**Fig. 3F, Fig. S7A**), and RRPC₂₂ (**Fig. 3G, Fig. S7A**). *PHF19* has been shown to negatively affect the expression of cell cycle inhibitors, therefore promoting proliferation.¹⁴ Taking into consideration the association of RRPC clusters with gain/amp1q, *TP53* mutations, and increased expression of *PHF19* (**Fig. S7B-D**), RRPC clusters and especially RRPC_{11,20,22} should be considered a high-risk subset of cells.

Fig. S7 Cell cycle phase enrichment by cluster. **A)** Total number of cells in each cluster stratified into cell cycle phase. **B)** Average *CKS1B* (CPM) expression for each cluster stratified by the contribution of cells from each phase. **C)** Average *PHF19* (CPM) expression for each cluster stratified by the contribution of cells from each phase. **D)** Average *MKI67* (CPM) expression for each cluster stratified by the contribution of cells from each phase.

Analysis of patient subclones indicates that those with Amp1q have increased expression of *PHF19*:

Clones 3 and 4 both had Amp1q in contrast to clones 1, 2, and 5 (**Fig. 6E**). *RBBP5*, *PBX1*, and *PHF19* were all significantly increased in expression (**Fig. 6F**) and had increased number of cells in G2M and S phases of cell cycle (**Fig. 6G**) in the clones with Gain/Amp1q. This may indicate that subclonal differences of Amp1q could affect *PHF19* expression through increased expression of the key TFs.

A second patient also contained subclonal heterogeneity represented by four clones that could be distinguished by clustering of RNA (**Fig. 6H**), ATAC (**Fig. 6I**), and WNN integration of RNA and ATAC (**Fig. 6J**). Based on WGS, this patient had Amp1q, and was hyperdiploid (**Fig. 6K**). Clone 4 had Amp1q (**Fig. 6L**), had significantly increased expression of *RBBP5*, *PBX1*, and *PHF19* compared to clones 1, 2, and 3 (**Fig. 6M**), and

had more cells in G2M and S phase of cell cycle (**Fig. 6N**). These examples demonstrate that even at the subclonal level there is likely regulation of *PHF19* by TFs on 1q, leading to high-risk disease.

Fig. 6 Subclonal differences in Gain/Amp1q. **A-C)** Patient sample 0661-533 clustering based on RNA (**A**), ATAC (**B**), WNN integrated RNA and ATAC (**C**). **D)** WGS based CNV profile for 0661-533. **E)** CNVs inferred from scRNA-seq data for 0661-533. **F)** Increased expression of *PBX1*, *RBBP5*, and *PHF19* in Amp1q clones from 0661-533. **G)** Increase in G2M and S phases of cell cycle for cells in Amp1q clones from 0661-533. **H-J)** Patient sample 0661-1043 clustering based on RNA (**H**), ATAC (**I**), WNN integrated RNA and ATAC (**J**). **K)** WGS based CNV profile. **L)** CNVs inferred from scRNA-seq data. **M)** Increased expression of *PBX1*, *RBBP5*, and *PHF19* in Amp1q clones. **N)** Increase in G2M and S phases of cell cycle for cells in Amp1q clones from 0661-1043. Fisher's exact test significance levels are: P<0.1: (-), P<0.05: *, P<0.01 (**), P<0.001 (***)

4. The authors should include details on quality control filtering performed and basic QC measures on the dataset, both for scRNA- and scATAC-sequencing. Marker genes/biological processes for cluster 22 seem to be enriched for ribosomal and mitochondrial RNAs, which can be sign of poor-quality cells. Therefore, some QC should be shown.

We appreciate this suggestion and have added an additional section and two additional tables in the supplementary materials about our QC process (see below).

SUPPLEMENTARY METHODS:

Single Cell QC

For the integration experiment, high quality cells were identified through a QC process. For each sample, the number of unique features and percentage of mitochondrial RNAs were plotted as violin plots. For each of the 49 samples these plots were evaluated to identify cutoffs based on the distribution of the data. A max number of unique RNAs, min number of unique RNAs, and a max percentage of mitochondrial RNAs were all set according to these initial violin plots (**Table S1**). In a similar fashion, cells were removed from the multiomic integration experiments using the ATAC data as well. Violin plots of the number of unique ATAC features, nuc signal, and TSS enrichment were plotted. Based on the distribution of each of these features a max and min cutoff value was set to remove outliers and low quality cells. A max number of unique ATAC features, min number of unique ATAC features, max nuc signal, min nuc signal, max TSS enrichment, and min TSS enrichment were all set according to these initial violin plots (**Table S2**).

Table S1 RNA QC metrics used for the preprocessing for the integration experiment and multiomic integration experiments.

Sample	max(RNA features)	min(RNA features)	max(mtRNA%)
Sample_0661_1053_5340777	4500	500	12
Sample_0661_1070_5347094	5000	500	10
Sample_0661_1127_5348553	4000	500	15
Sample_0661_1140_5350649	5500	500	20
Sample_0661_178_4153905	4000	500	20
Sample_0661_189_4829356	5000	500	10
Sample_0661_194_4156206	4500	500	10
Sample_0661_195_4172866	4000	500	15
Sample_0661_199_4680862	5500	500	20
Sample_0661_219_5167581	4500	500	15

Sample_0661_226_4678565	5500	500	10
Sample_0661_349_4156124	5000	500	10
Sample_0661_351_5350772	5000	500	15
Sample_0661_355_4680150	5500	500	10
Sample_0661_41_4308094	4500	500	8
Sample_0661_43_4046907	3500	500	15
Sample_0661_450_4408918	5000	500	20
Sample_0661_452_4306900	4500	500	8
Sample_0661_459_4312299	4500	500	10
Sample_0661_517_4835420	4000	500	12
Sample_0661_533_4835030	5000	500	12
Sample_0661_5_4680480	4000	500	15
Sample_0661_597_4677957	3500	500	20
Sample_0661_619_5690448	4500	500	15
Sample_0661_633_4447196	5000	500	20
Sample_0661_6_5961828	5500	500	15
Sample_0661_80_5965599	4500	500	15
Sample_0661_903_4835091	4500	500	12
Sample_0661_1008_4836785	5000	500	10
Sample_0661_1018_5339152	6000	500	15
Sample_0661_1021_5165969	4000	500	15
Sample_0661_1055_5340592	4000	500	8
Sample_0661_1059_5346970	5000	500	12
Sample_0661_1064_5346132	5000	500	15
Sample_0661_1092_5352736	4500	500	8
Sample_0661_455_4410289	2500	500	18
Sample_0661_634_4409412	4000	500	10
Sample_0661_1031_5167704	3800	500	10
Sample_0661_1034_5167024	4500	500	12
Sample_0661_1043_5167084	4500	500	18
Sample_0661_1066_5346256	4000	500	18
Sample_0661_1097_5357565	4500	500	20
Sample_0661_1132_5351658	3000	500	5
Sample_0661_1136_5352426	2500	500	12
Sample_0661_1146_5348025	4000	500	12
Sample_0661_1164_5698213	2000	500	12
Sample_0661_1184_5696042	3000	500	15
Sample_0661_1185_5696817	3000	500	8
Sample_0661_1274_5961619	3000	500	8

Table S2 ATAC QC metrics used for the preprocessing of the multiomic integration experiments

Sample	max(ATAC features)	min(ATAC features)	max(nuc signal)	min(nuc signal)	max(TSS enrichment)	min(TSS enrichment)
Sample_0661_533_4835030	10000	500	1.5	0.5	8	2
Sample_0661_189_4829356	5000	2000	1.2	0.7	8	3
Sample_0661_349_4156124	15000	500	1.3	0.8	8	3
Sample_0661_199_4680862	20000	500	1.5	0.7	7	3
Sample_0661_6_5_961828	10000	500	1.8	0.6	8	3
Sample_0661_43_4046907	8000	500	0.8	0.3	8	2
Sample_0661_80_5965599	10000	500	1.2	0.6	7	4
Sample_0661_178_4153905	5000	500	2.2	1	5	2.5
Sample_0661_194_4156206	18000	500	1	0.5	7	3
Sample_0661_219_5167581	8000	500	1.3	0.6	8	3
Sample_0661_226_4678565	9000	500	1.2	0.6	8	4
Sample_0661_355_4680150	18000	500	1.5	0.8	6	3
Sample_0661_450_4408918	18000	500	1.4	0.7	7	3
Sample_0661_633_4447196	15000	500	1.4	0.5	7	3
Sample_0661_634_4409412	15000	500	1	0.3	6	3
Sample_0661_101_8_5339152	15000	500	1.5	0.5	7	2
Sample_0661_109_7_5357565	5000	500	2.2	0.8	7	1.5
Sample_0661_5_4_680480	15000	500	1.2	0.7	7	3
Sample_0661_351_5350772	8000	500	1	0.4	7	3
Sample_0661_452_4306900	18000	500	1	0.5	7	3
Sample_0661_903_4835091	18000	500	1.2	0.7	6	3
Sample_0661_100_8_4836785	18000	500	1	0.5	6	3
Sample_0661_102_1_5165969	6000	500	3	1	7	1
Sample_0661_103_4_5167024	12000	500	1.5	0.5	7	3
Sample_0661_104_3_5167084	10000	500	1.5	0.5	8	3
Sample_0661_105_3_5340777	15000	500	1	0.4	7	3
Sample_0661_105_5_5340592	8000	500	2.3	0.8	6	2
Sample_0661_105_9_5346970	15000	500	1.2	0.4	7	2

Sample_0661_106 4_5346132	10000	500	1.7	0.7	8	2
Sample_0661_106 6_5346256	11000	500	1	0.2	7	2
Sample_0661_107 0_5347094	20000	500	1.3	0.7	5	2
Sample_0661_109 2_5352736	12000	500	1.2	0.6	7	2
Sample_0661_113 2_5351658	12000	500	1.2	0.5	7	3
Sample_0661_113 6_5352426	12000	500	1.3	0.7	7	2
Sample_0661_114 0_5350649	18000	500	1.5	0.5	6	3
Sample_0661_114 6_5348025	12000	500	1	0.5	6	2
Sample_0661_116 4_5698213	5000	500	2	0.8	6	2
Sample_0661_118 4_5696042	10000	500	0.9	0.3	8	2
Sample_0661_103 1_5167704	10000	500	1.5	0.6	7	2
Sample_0661_41_ 4308094	15000	500	1.8	0.8	5	3
Sample_0661_195 4172866	16000	500	1.3	0.5	7	2
Sample_0661_455 4410289	12000	500	1.5	0.7	9	3
Sample_0661_459 4312299	15000	500	1.5	0.7	7	2
Sample_0661_517 4835420	15000	500	1.6	0.7	6	3
Sample_0661_597 4677957	12000	500	1.3	0.7	8	2
Sample_0661_619 5690448	10000	500	1.2	0.4	8	2
Sample_0661_112 7_5348553	20000	500	1.4	0.7	6	2
Sample_0661_118 5_5696817	12000	500	1.3	0.5	8	3
Sample_0661_127 4_5961619	10000	500	1.2	0.6	6	3

5. Further details should be provided on some of the analyses. How was the association of particular cytogenetic events with clusters (shown in Table 3) determined? How were significant associations between clusters and covariates shown in Fig. 2D determined? How was statistical analysis performed (e.g. in Fig 2E-H)?

We thank the reviewer for bringing this up and have added additional information about the statistical analysis we performed. For Fig. 2D please see detailed description in the response to reviewer 2 comment d. The statistics in Fig. 2E-H were the same statistics described for Fig. 2D in the supplementary materials. See reviewer 2 comment d for a detailed description. Determining the significant associations is also described in this same section.

6. The role of Tp53 in this remains unclear.

This is a good point and we focused primarily on the 1q TFs for this study but are also very interested in how TP53 and genes on 1q interact. We can see from our data that the TP53 signal and 1q signal are not entirely dependent on each other. We would be interested in looking into how TP53 affect the proliferative signature as well in the future.

Minor comments:

1. Some of the figures are hard to read. In Fig. 2B, 2D, 3E-G the size of the legends should be increased.

We have increased the font sizes in Fig. 2B, 2D, and 3E-G. See reviewer 1 comment 2 for Fig. 2 and reviewer 1 comment 4 for Fig. 3.

2. The authors should make the size Venn diagram in Fig. 4C-E proportional to the number of genes.

This is a good point and we have updated Fig. 4C-E accordingly. Please see reviewer 2 comment j for the updated figure and figure legend.

3. The gene and promoter region in Fig. 5B should be labelled.

This is a great suggestion and we have added in the super-enhancer, promoter and promoter flank regions. See below.

Fig. 5 Association of the chromosome 1q TFs with *PHF19* expression. **A)** Location of TFs on chromosome 1q. **B)** PBX1 ChIP-seq peaks in promoter and enhancer region of *PHF19*

on chromosome 9. **C-D)** PBX1 ChIP-seq peak scores in MM1S (**C**) and U266 (**D**) myeloma cell lines with the percentile of each ChIP-seq peak marked. **E)** Differences in *PHF19* expression in MM1S cells treated with PBX1 shRNAs (P11 and P31). **F)** Differences in *PHF19* expression in U266 cells treated with PBX1 shRNAs (P11 and P31). **G)** Differential expression of *PBX1* and *PHF19* in RRPCs. Significance levels are: P<0.1: (·), P<0.05: *, P<0.01 (**), P<0.001 (***)

Reviewers' Comments:

Reviewer #1:

Remarks to the Author:

My initial review highlighted the significance of the submitted work to understanding the pathogenesis of proliferative MM. The authors have responded appropriately to exhaustive critiques, strengthening the statistical basis for their conclusions. The significance is not diminished.

Reviewer #2:

Remarks to the Author:

The authors did an excellent work in addressing questions from all 3 reviewers, of special importance was the use of COMMPASS database from MMRF to validate their findings.

However, there remain multiple questions that remain unaddressed. This reviewer believes the authors would greatly benefit from addressing them; not only they will make the article better, but will avoid having a high impact publication with possible fallacies:

1-Figure 2D remains problematic: I still don't think the claim from authors is accurate. One cannot claim that a particular cluster is enriched for mutations; it would be needed to very explicitly state that authors claim that samples that have a mutation are enriched for having cells in a particular cluster. Given the small percentage of cells in that RMM cluster (<15%), even in the enriched samples, the mutations cannot be from cells in those clusters, since mutations have high VAF (>20%) or do they?

2- Please check 2I; not sure if authors remembered to split gain and mutation, also * vs. . may be swapped between Normal and the two others? Ideally, we would have 4 violins, and statistical comparison would be between each and normal, then expected effect if "additive", then actual effect of the combination.

3-There is no reason for authors not use Cancer Hallmarks as well in their study. It should add minimum overhead, and these are curated for cancer, and well-accepted in community.

4-Please, make font of chr9 in Figure 5B larger, to avoid confusion with Chr1 in 5A.

5-Correlation between PHF19 and PBX1 (Pearson $r=0.21$) looks very weak. Could you please determine the percentile where it sits, compared to all gene pair correlations? This is specially important because it is also non-existent in our cohort, soon to be published with over 1,000 BM aspirates with RNA-seq, so this reviewer is just trying to help the authors not to make a claim that is not backed by real-world data.

6-Please, include FISH data. WGS does not replace FISH data, and it would be an additional source of data to confirm your findings. FISH has the advantage of containing data on percentage of cells positive or negative for each event.

7-The fact that the most important cluster is <15% of cells remains a problem. How can a small percentage of cells be responsible for clonal/sub-clonal cytogenetics and mutations? Also, if they are in RMM samples, how can they be responsible for tumor lack of response to therapy? Unless they communicate and mediate resistance of the other cells through paracrine loops, for example? Could you please address that in text?

8-This is merely a suggestion, not necessary for this article, but we have been working with scOmatic, which calculates mutations from scATAC-seq, in primary MM samples. While it has limitations, it is worth trying it. It makes mutation calls in cell clusters.

Reviewer #3:

Remarks to the Author:

The authors have made a comprehensive effort to address comments and questions, providing additional analyses and external datasets for validation. The manuscript benefits greatly from these improvements, enhancing the robustness and clarity of the findings. I have no further comments.

REVIEWER COMMENTS

We thank all of the reviewers for their insightful comments and general support for the article. We have attempted to thoroughly address all of the reviewer concerns and believe we have done so with additional clarification in the text, updated figures, and an additional RNA-seq validation experiment overexpressing PBX1 across two cell lines with different 1q CNV statuses. All of the manuscript authors have been very impressed with the quality of the reviews and with the amount of improvement made to the manuscript from this feedback. Our responses to reviewers are denoted with blue and direct quotes from the text are denoted with red.

Reviewer #1 (Remarks to the Author):

My initial review highlighted the significance of the submitted work to understanding the pathogenesis of proliferative MM. The authors have responded appropriately to exhaustive critiques, strengthening the statistical basis for their conclusions. The significance is not diminished.

We thank the reviewer for their enthusiastic support for the significance of our article and for their original insightful comments.

Reviewer #2 (Remarks to the Author):

The authors did an excellent work in addressing questions from all 3 reviewers, of special importance was the use of COMMPASS database from MMRF to validate their findings.

We thank the reviewer for spending the time to help us improve our article and believe that as a result our article is much more robust than before the first round of revisions.

However, there remain multiple questions that remain unaddressed. This reviewer believes the authors would greatly benefit from addressing them; not only they will make the article better, but will avoid having a high impact publication with possible fallacies:

1-Figure 2D remains problematic: I still don't think the claim from authors is accurate. One cannot claim that a particular cluster is enriched for mutations; it would be needed to very explicitly state that authors claim that samples that have a mutation are enriched for a having cells in a particular cluster. Given the small percentage of cells in that RMM cluster (<15%), even in the enriched samples, the mutations cannot be from cells in those clusters, since mutations have high VAF (>20%) or do they?

This is a good point and we have reworded this in the text using the phrasing provided by the reviewer to avoid inaccuracies. We have also added a section showing TP53 somatic mutations originally identified from WGS in the scATAC-seq reads from the RRPC₁₁ cells where we have read coverage. (lines: 254-256, 268-270, and 279-283)

METHODS

Using the previously defined clusters (**Fig. 2B**), we determined if samples in any disease stage, cytogenetic subgroup, or genetic subgroup are enriched for cells in any particular cluster (**Fig. 2D**).

Aside from a clear association of RRPCs with myeloma progression, there were also associations between samples with high-risk genomic events and their proportion of RRPCs (**Table S5, Fig. 2D**).

When the RRPC₁₁ fraction of cells was screened for Mut(*TP53*) in the scATAC-seq reads, i.e., variants originally identified by WGS, we were able to identify those variants in 3/5 of the samples with read coverage of the WGS variant loci (**Fig. S6**) and those RRPC₁₁ cells had an average Mut(*TP53*) VAF of 55% compared to 32% in all other cells (**Table S6**).

SUPPLEMENTARY MATERIALS

Fig. S6 Variants identified in WGS with coverage in the RRPC₁₁ fraction of cells scATAC-seq data for patients: 0661-1064 (A), 0661-1140 (B), 0661-43 (C), 0661-189 (D), 0661-450 (E).

Table S6 VAF table for *TP53* variants in scATAC-seq reads. *TP53* variants identified by WGS were screened in the scATAC-seq split into the RRPC₁₁ cells and cells from all other clusters. For each patient with a *TP53* variant identified by WGS, the VAF was calculated for that variant in the RRPC₁₁ cells and the other cells combined. *Denotes calculates made in only patient sample where both RRPC₁₁ and Other cells had read coverage.

Patient	Location	Variant Class	Variant Type	Mutation	dbSNP RS	RRPC ₁₁ VAF	Other cells VAF
0661-1064	Chr17:7,684,496	Intron	SNP	C->T	novel	100% (1/1)	60% (9/15)
0661-1140	Chr17:7,675,175	Nonsense	SNP	C->T	rs1206165503	75% (3/4)	20% (5/24)
0661-43	Chr17:7,687,374	Splice region	SNP	T->G	novel	0% (0/1)	67% (8/12)
0661-189	Chr17:7,675,088	Missense	SNP	C->T	rs28934578	0% (0/1)	67% (8/12)
0661-450	Chr17:7675088	Missense	SNP	C->T	rs28934578	100%	14% (1/7)
0661-41	Chr17:7,672,958	Intron	DEL		rs781465240		8% (1/13)
0661-178	Chr17: 7,673,609	Splice site	SNP	C->T	rs587781702		14% (2/14)
0661-349	Chr17: 7,674,291	Splice site	SNP	C->T	rs878854073		55% (5/9)
0661-349	Chr17:7,676,145-7,676,146	Frame shift	INS	->C	novel		13% (1/8)
0661-355	Chr17:7,663,150-7,663,151	Intron	INS	->TATT	rs370273514		
0661-597	Chr17: 7,669,977	Intron	SNP	C->T	rs1266557615		0% (0/5)
0661-1008	Chr17:7,674,902-7,674,919	In frame	DEL		novel		25% (2/8)
0661-1140	Chr17: 7674250	Missense	SNP	C->T	rs730882005		45% (5/11)
Mean						55%	32%
Median						75%	20%
Mean*						55%	32%
Median*						75%	20%

2- Please check 2I; not sure if authors remembered to split gain and mutation, also * vs. . may be swapped between Normal and the two others? Ideally, we would have 4 violins, and statistical comparison would be between each and normal, then expected effect if "additive", then actual effect of the combination.

This was a great suggestion from the initial review and sorry for any confusion. We followed the instructions outlined by reviewer 2 comment i: "Figure 2HG should have 4 columns (missing gain/Amp1q21) or at least two different symbols in middle violin." We used two different symbols, a diamond for Mut(*TP53*) and a circle for Gain/Amp1q(*CKS1B*) per the reviewer suggestion (see below).

We have since converted the symbol approach to the two violin approach in the latest manuscript (see below).

We appreciate the reviewer for catching this (* vs .). We double checked the statistics a second time and changed it to *. There were many comparisons in this figure and we do really appreciate the reviewer helping us to edit our manuscript.

Furthermore, we have updated the text to account for the updated figure (four violins in Fig. 2I) and included a concrete example of why the effect is additive. (lines: 283-289).

RESULTS

There was also an additive effect between Mut(*TP53*) status and Gain/Amp1q such that the number of these alterations was correlated with RRPC₁₁ proportion (PCC=0.47, P=0.001, **Fig. 2I**) and the RRPC₁₁ proportion in patients with both Gain/Amp1q and Mut(*TP53*) was equal to the RRPC₁₁ proportion in patients with Gain/Amp1q added to the RRPC₁₁ proportion in patients with Mut(*TP53*) accounting for the baseline RRPC₁₁ proportion in patients without either Gain/Amp1q or Mut(*TP53*) (**Fig. 2I**).

3-There is no reason for authors not use Cancer Hallmarks as well in their study. It should add minimum overhead, and these are curated for cancer, and well-accepted in community.

We have performed an additional GSEA analysis using the Cancer Hallmark ontologies to study which Cancer Hallmarks are enriched in the RRPC clusters. (lines: 323-326).

RESULTS

All RRPC clusters DEGs and DACs ($|\text{LogFC}| > 0.4$, $P < 0.05$) were also enriched (GSEA FDR < 0.25) for cancer Hallmarks like E2F targets, G2M checkpoint, and immune signaling pathways (**Table S12-S14**).

SUPPLEMENTARY MATERIALS

Table S12 Cancer Hallmarks enriched for DEGs and DACs from RRPC₁₁

CANCER HALLMARK	ES	NES	P	FDR	size	DEG type
E2F_TARGETS	0.70	3.16	0.00E+00	0.00E+00	188	RNA
G2M_CHECKPOINT	0.69	3.06	0.00E+00	0.00E+00	149	RNA
MITOTIC_SPINDLE	0.55	2.36	0.00E+00	0.00E+00	119	RNA
SPERMATOGENESIS	0.54	2.19	0.00E+00	0.00E+00	71	RNA
TNFA_SIGNALING_VIA_NFKB	-0.41	-2.70	0.00E+00	0.00E+00	76	RNA
TNFA_SIGNALING_VIA_NFKB	-0.63	-3.63	0.00E+00	0.00E+00	22	ATAC
KRAS_SIGNALING_UP	-0.51	-2.74	0.00E+00	3.16E-04	19	ATAC
HYPOXIA	-0.59	-2.46	0.00E+00	3.16E-03	11	ATAC
UV_RESPONSE_UP	-0.80	-2.22	8.63E-04	8.55E-03	5	ATAC
IL2_STAT5_SIGNALING	-0.39	-1.94	1.31E-02	3.20E-02	17	ATAC
ESTROGEN_RESPONSE_LATE	-0.34	-1.68	2.41E-02	8.04E-02	16	ATAC
OXIDATIVE_PHOSPHORYLATION	-0.23	-1.78	0.00E+00	8.29E-02	137	RNA
MYOGENESIS	-0.34	-1.70	1.78E-02	8.86E-02	16	ATAC
ESTROGEN_RESPONSE_EARLY	-0.33	-1.57	5.13E-02	1.17E-01	15	ATAC
DNA_REPAIR	0.36	1.53	1.12E-02	1.22E-01	99	RNA
PANCREAS_BETA_CELLS	-0.47	-1.53	6.30E-02	1.23E-01	7	ATAC
HYPOXIA	-0.22	-1.61	0.00E+00	1.62E-01	97	RNA
IL6_JAK_STAT3_SIGNALING	-0.42	-1.38	1.03E-01	1.89E-01	7	ATAC

INFLAMMATORY_RESPONSE	-0.29	-1.39	1.09E-01	2.01E-01	15	ATAC
-------	-------	----------	----------	----	------

Table S13 Cancer Hallmarks enriched for DEGs and DACs from RRPC₂₀

CANCER HALLMARK	ES	NES	P	FDR	size	DEG type
E2F_TARGETS	0.63	2.54	0.00E+00	0.00E+00	165	RNA
G2M_CHECKPOINT	0.69	2.76	0.00E+00	0.00E+00	146	RNA
HEME_METABOLISM	0.50	1.95	0.00E+00	1.06E-03	95	RNA
INTERFERON_ALPHA_RESPONSE	-0.22	-1.42	1.28E-02	2.41E-01	56	RNA
MITOTIC_SPINDLE	0.60	2.38	0.00E+00	0.00E+00	116	RNA
OXIDATIVE_PHOSPHORYLATION	-0.23	-1.72	0.00E+00	8.82E-02	108	RNA
SPERMATOGENESIS	0.52	1.94	0.00E+00	1.05E-03	63	RNA
TNFA_SIGNALING_VIA_NFKB	-0.34	-2.29	0.00E+00	4.77E-03	67	RNA
IL2_STAT5_SIGNALING	-0.44	-1.66	3.01E-02	1.19E-01	12	ATAC
INFLAMMATORY_RESPONSE	-0.43	-1.98	4.84E-03	2.59E-02	20	ATAC
TNFA_SIGNALING_VIA_NFKB	-0.62	-2.39	0.00E+00	1.53E-03	13	ATAC

Table S14 Cancer Hallmarks enriched for DEGs and DACs from RRPC₂₂

CANCER HALLMARK	ES	NES	P	FDR	size	DEG type
APICAL_SURFACE	-0.36	-1.36	1.11E-01	2.30E-01	16	RNA
COAGULATION	0.41	1.70	1.01E-02	2.42E-02	45	RNA
DNA_REPAIR	0.35	1.70	4.18E-03	2.56E-02	105	RNA
E2F_TARGETS	0.55	2.86	0.00E+00	0.00E+00	162	RNA
EPITHELIAL_MESENCHYMAL_TRANSITION	-0.26	-1.57	1.58E-02	8.58E-02	79	RNA
ESTROGEN_RESPONSE_EARLY	-0.34	-2.00	0.00E+00	1.28E-02	71	RNA
G2M_CHECKPOINT	0.65	3.34	0.00E+00	0.00E+00	142	RNA
GLYCOLYSIS	0.31	1.51	1.57E-02	9.11E-02	103	RNA
INFLAMMATORY_RESPONSE	-0.27	-1.57	2.00E-02	1.02E-01	69	RNA
INTERFERON_ALPHA_RESPONSE	0.35	1.57	1.82E-02	6.23E-02	64	RNA
MITOTIC_SPINDLE	0.44	2.20	0.00E+00	1.34E-04	120	RNA
MTORC1_SIGNALING	0.30	1.50	1.25E-02	9.13E-02	131	RNA
MYC_TARGETS_V1	0.35	1.84	0.00E+00	7.80E-03	169	RNA
MYC_TARGETS_V2	0.46	1.91	0.00E+00	4.49E-03	46	RNA
OXIDATIVE_PHOSPHORYLATION	0.35	1.79	0.00E+00	1.19E-02	159	RNA
SPERMATOGENESIS	0.53	2.31	0.00E+00	0.00E+00	56	RNA
TNFA_SIGNALING_VIA_NFKB	-0.31	-1.88	0.00E+00	1.98E-02	87	RNA
UV_RESPONSE_DN	-0.40	-2.47	0.00E+00	0.00E+00	78	RNA
DNA_REPAIR	0.65	2.70	0.00E+00	1.21E-03	15	ATAC
E2F_TARGETS	0.55	2.31	1.14E-03	4.15E-03	16	ATAC
G2M_CHECKPOINT	0.57	2.17	3.00E-03	7.13E-03	12	ATAC
INTERFERON_ALPHA_RESPONSE	0.51	2.12	1.10E-03	6.03E-03	16	ATAC
INTERFERON_GAMMA_RESPONSE	0.40	2.14	0.00E+00	6.59E-03	29	ATAC
MYC_TARGETS_V1	0.56	1.86	1.30E-02	2.95E-02	9	ATAC

OXIDATIVE_PHOSPHORYLATION	0.54	2.31	3.29E-03	6.03E-03	16	ATAC
UV_RESPONSE_DN	0.35	1.33	1.61E-01	2.48E-01	13	ATAC

4-Please, make font of chr9 in Figure 5B larger, to avoid confusion with Chr1 in 5A. We thank the reviewer for noticing this and agree that this could be confusing to a reader. We have since increased the font size in Fig. 5B.

5-Correlation between PHF19 and PBX1 (Pearson $r=0.21$) looks very weak. Could you please determine the percentile where it sits, compared to all gene pair correlations? This is specially important because it is also non-existent in our cohort, soon to be published with over 1,000 BM aspirates with RNA-seq, so this reviewer is just trying to help the authors not to make a claim that is not backed by real-world data.

We appreciate the reviewers concern and have added an additional validation experiment to our study. In this experiment we overexpress PBX1 in two myeloma cell lines one with 1q gain/amplification (MM1S) and one without 1q gain/amplification (PCM6). After overexpressing PBX1 in each of these cell lines, we compared the expression profile to the cell lines with an empty vector. In short, we see the opposite effect on PHF19 expression (Fig. 5H-K) than the knockdown experiment that we previously included (Fig. 5 E-F). We see a larger effect size in PCM6 which does not already have multiple copies of 1q than in MM1S which does. In both cases *PHF19* has increased expression ($\alpha=0.1$, i.e. PCM6 $P=0.011$ and MM1S $P=0.089$). We have achieved quite significant results given a sample size of only two per group, that align with our previous results, and whose effect size is affected by the existence of gained/amplified 1q in the cell lines. Additionally, we calculated the percentile of the PBX1/PHF19 correlation and found that it was in the 83rd percentile of *PBX1* correlations. (lines: 224-236, 392-400, and 405-408).

METHODS

Functional evaluation of PBX1 TF activity on PHF19 using RNA-seq

Previous studies already conducted knockdown experiments of *PBX1* using two shRNAs in the cell lines MM1S and U266 where global expression was measured using RNA-seq. We used a one-sided t-test in each cell line to test whether *PHF19* expression was reduced in each *PBX1* shRNA group.

To further strengthen these relationships, we also performed a *PBX1* overexpression experiment in two myeloma cell lines, one with gain/amplification 1q (MM1S) and one without gain/amplification 1q (PCM6). After these cell lines were grown and processed (**Supplementary Materials**) with either *PBX1* overexpression (oePBX1) or empty vector (eVec) control, the expression of *PHF19* was compared using a one-sided t-test. The global gene expression changes were also analyzed with EdgeR³³ where a gene was considered a DEG if $|\text{Log}_2\text{FC}| > 1$ and FDR of $< 1E-5$.

RESULTS

When myeloma cell lines overexpressed *PBX1*, we saw the opposite effect of the knockdown of *PBX1* on *PHF19* expression (**Fig. 5H-K**). When *PBX1* was overexpressed in *PCM6* (**Fig. 5H**), *PHF19* expression is significantly increased (**Fig. 5I**)

and when *PBX1* was overexpressed in MM1S (Fig. 5J), *PHF19* expression was also increased ($P < 0.1$, Fig. 5K). The effect size from *PBX1* overexpression was much larger in the PCM6 cells ($\text{Log}_2\text{FC} = 0.75$) than the MM1S cells ($\text{Log}_2\text{FC} = 0.06$) corresponding inversely to their 1q CNV statuses. We also see notable up-regulation of cell cycle genes like *TOP2A* (Fig. S11A+B, Table S15+S16) and *MKI67* (Fig. S11B, Table S16) and down-regulation of immune related genes like *CXCL10* (Fig. S11A, Table S15).

When we compared the correlation between *PBX1* and *PHF19* in the cohort of NDMM patients from the MMRF CoMMpass study, we again identified a significant correlation ($\text{PCC} = 0.21$, $P < 0.001$, $> 83.30\%$ *PBX1* PCCs).

Fig. 5 Association of the chromosome 1q TFs with *PHF19* expression. **A)** Location of TFs on chromosome 1q. **B)** *PBX1* ChIP-seq peaks in promoter and enhancer region of *PHF19* on chromosome 9. **C-D)** *PBX1* ChIP-seq peak scores in MM1S (**C**) and U266 (**D**) myeloma cell lines with the percentile of each ChIP-seq peak marked. **E)** Differences in *PHF19* expression in MM1S cells treated with *PBX1* shRNAs (P11 and

P31). **F**) Differences in *PHF19* expression in U266 cells treated with *PBX1* shRNAs (P11 and P31). **G**) Differential expression of *PBX1* and *PHF19* in RRPCs. **H**) *PBX1* expression in PCM6 eVec cells compared to PCM6 cells with a vector overexpressing *PBX1* (oe*PBX1*). **I**) *PHF19* expression in PCM6 eVec cells compared to PCM6 oe*PBX1* cells. **J**) *PBX1* expression in MM1S eVec cells compared to MM1S oe*PBX1* cells. **K**) *PHF19* expression in MM1S eVec cells compared to MM1S oe*PBX1* cells. Significance levels are: $P < 0.1$: (·), $P < 0.05$: *, $P < 0.01$ (**), $P < 0.001$ (***)

SUPPLEMENTARY MATERIALS

Analysis of myeloma cell line RNA-seq data

Knockdown experiments were performed for MM1S and U266 cell lines that were utilized to study the effects of *PBX1* expression downregulation on *PHF19* expression. Processed data were downloaded from GSE165060. For complete methods please see corresponding publication.

Overexpression experiments were performed on myeloma cell lines to study the effect of *PBX1* expression up-regulation on *PHF19* expression. The vector pLenti-C-Myc-DDK-P2A-puro (Origene) with either the *PBX1* transcript (NM_002585) or empty vector was transduced into PCM6 and MM1-S multiple myeloma cell lines. Virus was produced from HEK 293T cells with VSV-G and PSPAX2 helper plasmids. After transduction, cells underwent selection with puromycin (0.5 $\mu\text{g}/\text{mL}$). Resistant cells from both *PBX1* and empty vector transduced cells were grown and subsequently RNA was extracted in duplicate using the RNeasy Mini kit (Qiagen). 100 nanograms of total RNA was used for library preparation utilizing the Illumina Stranded mRNA Prep kit. 150 bp paired-end reads were generated on an Illumina NovaSeq 6000 sequencer with a target depth of 50M reads. Reads were aligned to the GRCh38 primary assembly with GENCODE comprehensive gene annotation for reference chromosomes (release 45) using STAR (v2.7.11a). The aligned reads were then quantified to their respective genes using featureCounts (v2.0.6).

Fig. S11 Differential expression results from oePBX1 compared to eVec in PCM6 **(A)** and MM1S **(B)** cell lines. Top 10 up- and down-regulated DEGs (based on FDR then $|\text{Log}_2\text{FC}|$) are labeled in the volcano plots. DEGs ($|\text{Log}_2\text{FC}| > 1$ and BH-FDR $< 1e-5$) are marked in red.

Table S15 Top 25 up- and down-regulated DEGs (based on FDR then $|\text{Log}_2\text{FC}|$) in PCM6 cell line for oePBX1 and eVec treated cells. The remaining significant DEGs at ($|\text{Log}_2\text{FC}| > 1$ and BH-FDR $< 1e-5$) are can be found in **Supplementary File 1**.

Gene	Log ₂ FC	P	FDR
PBX1	15.51	0.00E+00	0.00E+00
PRR11	3.92	0.00E+00	0.00E+00
FCRLA	3.51	0.00E+00	0.00E+00
PLK1	3.24	0.00E+00	0.00E+00
MT-CO3	3.04	0.00E+00	0.00E+00
TOP2A	2.97	0.00E+00	0.00E+00
ASPM	2.95	0.00E+00	0.00E+00
MT-ATP8	2.78	0.00E+00	0.00E+00
MT-ATP6	2.78	0.00E+00	0.00E+00
MTATP6P1	2.73	0.00E+00	0.00E+00
MT-CYB	2.66	0.00E+00	0.00E+00
MT-ND6	2.59	0.00E+00	0.00E+00
MT-ND4	2.57	9.30E-314	8.58E-311
MT-CO1	2.34	1.32E-310	1.18E-307
CENPF	2.90	3.73E-297	3.16E-294
GTSE1	3.12	9.92E-294	7.95E-291
MT-ND3	2.42	4.80E-289	3.66E-286
FOXM1	2.54	1.27E-287	9.22E-285
MT-ND5	2.49	1.90E-287	1.35E-284
NMU	3.80	1.39E-285	9.44E-283
SPAG5	2.57	1.48E-285	9.80E-283
COL24A1	3.21	3.03E-273	1.84E-270
NUSAP1	2.30	4.97E-252	2.44E-249
HMGB2	2.26	9.49E-250	4.52E-247
TK1	2.81	8.25E-241	3.70E-238
CXCL10	-8.26	0.00E+00	0.00E+00
ART3	-8.03	0.00E+00	0.00E+00
GBP4	-6.00	0.00E+00	0.00E+00
CCL2	-5.41	0.00E+00	0.00E+00
RSAD2	-5.27	0.00E+00	0.00E+00
IFIT2	-4.77	0.00E+00	0.00E+00

MSR1	-4.35	0.00E+00	0.00E+00
DAB2	-3.83	0.00E+00	0.00E+00
RIGI	-3.58	0.00E+00	0.00E+00
APOBEC3A	-3.55	0.00E+00	0.00E+00
TNFSF13B	-3.19	0.00E+00	0.00E+00
CCDC80	-3.04	0.00E+00	0.00E+00
TEX29	-2.99	0.00E+00	0.00E+00
CCDC146	-2.94	0.00E+00	0.00E+00
GBP1	-2.92	0.00E+00	0.00E+00
COL27A1	-2.77	0.00E+00	0.00E+00
OASL	-2.66	0.00E+00	0.00E+00
TNFSF10	-2.66	5.04E-322	5.12E-319
RSP01	-2.99	2.45E-320	2.41E-317
FGL2	-3.03	6.46E-318	6.15E-315
LINC02574	-2.44	2.71E-301	2.36E-298
IFI6	-2.35	8.13E-294	6.69E-291
LINC00364	-2.40	2.53E-292	1.98E-289
RASGEF1B	-2.67	8.02E-288	5.96E-285
C3AR1	-3.52	3.93E-286	2.73E-283

Table S16 Top 25 up- and down-regulated DEGs (based on FDR then $|\text{Log}_2\text{FC}|$) in MM1S cell line for oePBX1 and eVec treated cells. The remaining significant DEGs at ($|\text{Log}_2\text{FC}| > 1$ and BH-FDR $< 1e-5$) are can be found in **Supplementary File 2**.

	log2FC	PValue	FDR
PBX1	10.26	0.00E+00	0.00E+00
MKI67	4.23	1.79E-217	1.07E-214
FAM111B	2.12	6.53E-197	3.62E-194
STMN1	2.26	2.35E-163	1.14E-160
RRM2	4.52	5.91E-157	2.73E-154
MCM4	1.17	4.75E-141	2.04E-138
TOP2A	4.58	1.37E-122	4.80E-120
MCM2	1.22	1.13E-118	3.79E-116
DTL	1.92	5.29E-115	1.73E-112
MCM5	1.06	2.01E-73	4.71E-71
RASSF6	1.20	7.81E-59	1.54E-56
CDC6	2.11	9.02E-59	1.75E-56
E2F1	1.56	4.69E-54	8.51E-52
ASPM	3.98	4.34E-51	7.52E-49
UHRF1	2.01	1.41E-46	2.28E-44

CENPF	1.92	1.16E-42	1.75E-40
ZNF367	1.38	2.05E-38	2.79E-36
CDT1	1.30	2.57E-38	3.48E-36
SHCBP1	2.30	6.44E-38	8.54E-36
PCLAF	2.74	1.32E-37	1.74E-35
UBE2C	4.69	1.81E-37	2.35E-35
ZWINT	2.51	6.28E-37	7.98E-35
CCNB2	4.01	2.26E-35	2.69E-33
TROAP	4.73	6.21E-34	7.04E-32
MCM10	3.37	6.43E-33	7.10E-31
LINC00529	-8.63	0.00E+00	0.00E+00
MTUS2	-7.78	0.00E+00	0.00E+00
GRB10	-6.16	0.00E+00	0.00E+00
STMN3	-4.42	0.00E+00	0.00E+00
NFATC1	-4.34	0.00E+00	0.00E+00
LINC02337	-3.97	0.00E+00	0.00E+00
TAS1R1	-3.73	0.00E+00	0.00E+00
GLI2	-3.72	0.00E+00	0.00E+00
LINC01795	-3.65	0.00E+00	0.00E+00
SPIB	-3.65	0.00E+00	0.00E+00
LINC00276	-3.64	0.00E+00	0.00E+00
RSP01	-3.61	0.00E+00	0.00E+00
GULP1	-3.56	0.00E+00	0.00E+00
LINC01090	-3.55	0.00E+00	0.00E+00
RIMBP2	-3.53	0.00E+00	0.00E+00
TEX29	-3.50	0.00E+00	0.00E+00
C19orf73	-3.48	0.00E+00	0.00E+00
SLC18A1	-3.47	0.00E+00	0.00E+00
SPANXA2-OT1	-3.38	0.00E+00	0.00E+00
RAD51B	-3.32	0.00E+00	0.00E+00
RALYL	-3.30	0.00E+00	0.00E+00
SLIT3	-3.28	0.00E+00	0.00E+00
OXCT2P1	-3.25	0.00E+00	0.00E+00
IGSF3P2	-3.24	0.00E+00	0.00E+00
OXCT2	-3.23	0.00E+00	0.00E+00

6-Please, include FISH data. WGS does not replace FISH data, and it would be an additional source of data to confirm your findings. FISH has the advantage of containing data on percentage of cells positive or negative for each event.

We completely understand the reviewers point and we really appreciate the fact that this could strengthen the study. However, there seems to be a little confusion on this point about the data and samples that we have and we hope that a more thorough explanation will help. In short, there are some feasibility issues in our specific case due to the way our patients were enrolled, samples biobanked, and samples processed that significantly affect the utility of this approach and the practicality of performing these analyses. Clinical FISH was performed on unselected cells and so cannot be used to determine percentage of plasma cells with any abnormality. Since clinical FISH was performed, contamination with normal cells will not give you correct percentages. For many of these samples, we no longer have aliquots of viable cells remaining in our biobank due to the extensive amount of multiomics we have been performing on these samples in this study and many others. For these reasons, it is not possible to retrospectively perform FISH and the FISH we do have will not give us accurate estimates of the abnormalities. Furthermore, acquiring new samples from many of these patients is not possible given many were enrolled years ago. For patients still coming to the clinic, performing more bone marrow aspirates to get selected FISH data may affect these patient's quality of life unnecessarily especially given we already have WGS data that may not perfectly replace FISH but does provide much of the same information.

Despite not being an exact replacement for FISH, we do want to point out that WGS is highly concordant to FISH as outlined by multiple studies, many of which in myeloma (10.1016/j.cancer.2020.01.001, 10.1182/blood-2021-152412, [10.1182/blood-2019-125145, <https://doi.org/10.1002/jha2.276>], 10.1056/NEJMoa2024534, 10.1016/j.jmoldx.2021.04.011, 10.1158/1078-0432.CCR-21-3695).

7-The fact that the most important cluster is <15% of cells remains a problem. How can a small percentage of cells be responsible for clonal/sub-clonal cytogenetics and mutations? Also, if they are in RMM samples, how can they be responsible for tumor lack of response to therapy? Unless they communicate and mediate resistance of the other cells through paracrine loops, for example?

We agree with the reviewer that we needed further explanation of this point and have added this in the discussion. (lines: 446-456)

It is intriguing that these high-risk RRPCs only make up <17% of an RRMM patient's myeloma cells. However, this is not to say that they do not have an outsized effect on patient survival or interact with other non-RRPCs to promote disease. In other cancers, intra-tumor heterogeneity leads to interactions between subclones, through cell signaling, leading to diffuse infiltration and complicating disease management³⁶. Positive interactions from one subclone can lead to the population as a whole benefiting, resulting in commensalism³⁷. Examples of such benefits could be secreted factors, weakening of the immune system, or stimulation of the bone marrow to provide a protective niche³⁶⁻³⁹. Determining the function and interactions of small high-risk clones with both other myeloma cells and the microenvironment will increase our understanding of resistance and relapsed disease.

8-This is merely a suggestion, not necessary for this article, but we have been working with scOmatic, which calculates mutations from scATAC-seq, in primary MM samples. While it has limitations, it is worth trying it. It makes mutation calls in cell clusters.

This is an excellent suggestion and we have actually been looking for a tool to do this for some time. We will be sure to include this in our subsequent work.

Reviewer #3 (Remarks to the Author):

The authors have made a comprehensive effort to address comments and questions, providing additional analyses and external datasets for validation. The manuscript benefits greatly from these improvements, enhancing the robustness and clarity of the findings. I have no further comments.

We thank the reviewer for their feedback and support.

Reviewers' Comments:

Reviewer #2:

Remarks to the Author:

I would like to thank the authors for clarifying the questions I had in the previous reviews. I have learned much for this extraordinary article.

I believe the authors have addressed all questions, and this work will be a great credit to this Journal, and move the MM field forward.